# *In silico* design of novel recombinant antigens containing immunologically relevant regions of wild-type and escape mutant variants of HBsAg

Yeshwas Abite Workneh[1,2,3]*, Desye Melese Sisay[2,4], Abebaw Fekadu[2,5],
Abraham Tesfaye Bika[2], Alemu Tekewe Mogus[2], Tesfaye Sisay Tessema[1]

1 Biotechnology Research Center, Addis Ababa University, Addis Ababa, Ethiopia, 2 CDT-Africa, College of Health Sciences, Addis Ababa University, Addis Ababa, Ethiopia, 3 Department of Pathology, College of Health Sciences, Addis Ababa University, Addis Ababa, Ethiopia, 4 Department of Biotechnology, College of Natural Sciences, Wollo University, Dessie, Amhara, Ethiopia, 5 Global Health & Infection Department, Brighton and Sussex Medical School, Brighton, UK

* yeshwas.abite@aau.edu.et

## Abstract

Hepatitis B virus (HBV) contributes substantially to liver cancer, related mortality, and liver transplantation worldwide. The small hepatitis B surface antigen (HBsAg), particularly its major hydrophilic region (MHR) and the "a" determinant, is the primary target of serological diagnostics. However, escape mutant amino acid variants (EMAVs) within this region may reduce diagnostic specificity and sensitivity. In this study, publicly available HBsAg sequences were analyzed to determine the prevalence of EMAVs circulating in Ethiopia. We computationally designed three region-specific recombinant antigens (MeRPYS1, MeRPYS2, and MeRPYS3) by incorporating both wild-type and prevalent EMAV sequences. Linear and conformational B-cell epitopes, as well as T helper cell epitopes, were predicted for each antigen. Homology analyses were also performed to assess similarity to host proteins. Secondary and tertiary structures of the antigens were predicted to generate theoretical molecular models. Molecular docking analyses were performed to explore putative interaction patterns between each designed antigen and an anti-HBsAg-specific antibody. The predicted antigen–antibody complexes were further examined using molecular dynamics (MD) simulations to assess their theoretical stability and behavior over time. The resulting simulations provide predictive computational insights into possible antigenic features and interaction tendencies of the designed constructs. These findings are intended to generate testable hypotheses and should be interpreted cautiously, as the study is limited to *in silico* analyses and requires experimental validation.

## Introduction

Viral hepatitis remains a significant global health burden, causing deaths primarily due to cirrhosis and liver cancer [1]. In 2015, viral hepatitis accounted for 1.34 million

**Data availability statement:** All relevant data are within the manuscript and supporting information.

**Funding:** The author(s) received no specific funding for this work.

**Competing interests:** The authors have declared that no competing interests exist.

deaths, while in 2022 this figure was estimated at 1.3 million. These mortality estimates are comparable to those attributed to tuberculosis and exceed deaths caused by HIV. A majority (over 82%) of these deaths were attributed to chronic HBV infection [2,3].

In 2022, 254 million people were living with HBV, with 1.2 million new infections reported that year. Despite this substantial burden, only 13% of individuals with chronic HBV were diagnosed, and merely 3% of them received treatment [3]. These statistics highlight major gaps in HBV diagnosis and treatment.

HBV is a compact virus with a partially double-stranded DNA genome and is a member of the *Hepadnaviridae* family. First identified in 1965, HBV continues to be a major cause of both acute and chronic hepatitis. It spreads primarily through percutaneous, sexual, and parenteral transmission routes, targeting hepatocytes where it replicates and induces liver damage [4].

The genome of HBV is approximately 3.2 kb in length, consisting of relaxed-circular DNA with four overlapping open reading frames (ORFs) coding for essential viral proteins, C, P, S, and X. The S ORF produces the envelope proteins L-HBsAg, M-HBsAg, and S-HBsAg, with S-HBsAg being the most abundantly expressed protein. S-HBsAg forms both the viral envelope and sub-viral particles, which are present in much greater quantities than the infectious virus itself. Due to its abundance, S-HBsAg is a key focus in diagnostic, vaccine and therapeutic studies [5,6].

The S-HBsAg protein comprises 226 amino acids and features critical regions like the MHR and the conserved "a" determinant [7]. These critical regions play a vital role in immune recognition of the virus and are targets to the development of diagnostic assays. However, mutations within these regions can reduce the accuracy of diagnostic tools by allowing the virus to evade detection. Mutations such as I110L, A128V, and Y134F are common examples of diagnostic EMAVs that have been associated with reduced sensitivity and specificity of immunological assays [8,9].

HBV is divided into nine genotypes, labeled A through I. HBV genotypes are defined by amino acid variation within the MHR and show distinct geographic distributions [10,11]. In Ethiopia, genotypes A and D are the most prevalent, alongside occasional reports of genotypes E, C, and F [12–17].

The prevalence of HBV in Ethiopia is estimated to range from 5% to 8%. This prevalence classifies Ethiopia as a high-burden setting and represents a major public health concern [18].

In Ethiopia, commonly used HBV diagnostic tools [19] are imported from abroad. Despite their widespread use, the diagnostic tools often show variable performance. For example, a WHO-prequalified diagnostic kit tested in Ethiopia showed a sensitivity of around 80% which was lower than the sensitivity values reported under controlled validation settings [20,21].

To address diagnostic challenges arising from the genetic diversity and mutation profile of HBV in Ethiopia, this study applied a region-specific computational strategy for HBsAg-based antigen design. Publicly available HBsAg amino acid sequences were analyzed to identify predominant circulating genotypes and prevalent EMAVs within the MHR and the 'a' determinant. Based on these analyses, recombinant

antigen constructs incorporating immunologically relevant wild-type and EMAV-derived regions were designed *in silico*. The resulting constructs were further characterized using computational tools to generate predictive insights into their sequence features, structural properties, and theoretical interaction patterns, including putative compatibility with bacterial expression systems. The novelty of this work lies in the systematic prioritization of Ethiopia-specific HBV sequence variation to inform hypothesis-driven antigen design. These findings are intended to guide subsequent experimental validation rather than to demonstrate confirmed antigenicity, diagnostic performance, or suitability for antibody production.

## Methods and materials

### Identification of HBsAg EMAVs

The amino acid sequences of HBsAg from HBV genotypes circulating in Ethiopia were systematically analyzed using pooled sequence data. All publicly available HBsAg amino acid sequences deposited in the National Center for Biotechnology Information (NCBI) database up to August 2025 were retrieved. A total of 682 HBsAg sequences, each derived from a unique individual, were collected using their corresponding accession numbers. These sequences were distributed across five accession ranges: KP310929–KP311299 (371 sequences), KT367571–KT367731 (161 sequences), MF169791–MF169875 (85 sequences), OP432487–OP432532 (46 sequences), and OL630698–OL630716 (19 sequences).

Accession numbers KT367574 and OP432510 were excluded prior to analysis. Accession KT367574 represented the only genotype E sequence in the dataset and was excluded due to insufficient representation for genotype-level comparative analysis. Accession OP432510 was excluded because the corresponding HBsAg protein sequence was not deposited in the database and was annotated as nonfunctional due to mutation. In addition, ten isolates (KT367572, KT367612, KT367614, KT367620, KT367631, KT367643, KT367684, KT367687, OP432515, and OL630711) containing at least one unidentified amino acid (X) within the "a" determinant region were excluded to avoid ambiguity in mutation analysis. Following these exclusions, 670 HBsAg sequences were retained for further analysis.

The retained sequences were categorized by genotype (A or D) and independently aligned against genotype-specific reference sequences using BioEdit Sequence Alignment Editor (version 7.2.5). HBV isolates AY934770 and AB188243 served as reference sequences for genotypes A and D, respectively. EMAVs were identified by aligning the MHR of each sequence against its corresponding reference sequence and recording amino acid substitutions within the defined region.

### Statistical analysis

Genotype distribution and mutation frequencies were calculated as proportions based on the total number of HBsAg sequences with complete MHR data. Ninety-five percent confidence intervals (95% CIs) for proportions were estimated using the Wilson score method without continuity correction.

To assess whether the prevalence of EMAVs differed significantly between HBV genotypes A and D, a two-proportion z-test was performed. The proportion of sequences harboring at least one EMAV in the MHR was calculated for genotype A as:

$$pA \ = \ xA/nA$$

It was calculated for genotype D as:

$$pD \ = \ xD/nD$$

Where $xA$ and $xD$ represent the number of sequences with ≥1 MHR mutation and $nA$ and $nD$ denote the total number of sequences analyzed for each genotype, respectively.

The pooled proportion (Ppooled) was calculated as:

$$P\text{pooled} = (xA + xD) / (nA + nD)$$

The standard error (SE) of the difference in proportions was computed using:

$$SE = \sqrt{P\text{pooled}\,(1 - P\text{pooled})\,(1/nA + 1/nD)}$$

The z-score (z) was then calculated as:

$$z = (pD - pA)/SE$$

P-values < 0.05 were considered indicative of statistical significance. Effect sizes were also reported as the absolute difference in proportions ($\Delta p = |pD - pA|$).

All eligible sequences available up to the date of data retrieval were included in the analysis, except those explicitly excluded based on predefined criteria, to minimize selection bias.

## Design of recombinant proteins

Three recombinant protein constructs (MeRPYS1, MeRPYS2, and MeRPYS3) were designed using a computational workflow. MeRPYS1 and MeRPYS2 were composed of ten tandem repeats of the HBsAg "a" determinant derived from HBV genotypes A and D. The relative proportions of genotype A– and genotype D–derived sequences differed between the two constructs and were informed by genotype prevalence and mutation frequency observed in the Ethiopian sequence dataset. Selected diagnostic EMAVs were incorporated into specific "a" determinant segments, while remaining segments consisted of wild-type sequences. Individual "a" determinant segments were separated by a flexible glycine–serine (GSGSG) linker. A fusion-tag was included at the N-terminus of each construct, and a hexa-his-tag was appended at the C-terminus.

MeRPYS3 was designed by incorporating two copies of the wild-type MHR from genotype A and two copies from genotype D. The MHR segments were separated by the GSGSG linker. A combination of fusion-tags, including a Strep-tag II, a hexa-his-tag, and a Twin-Strep-tag, was incorporated at the N-terminus of the MeRPYS3 construct.

## Protein homology test

Potential similarity between the designed recombinant proteins (MeRPYS1, MeRPYS2, and MeRPYS3) and host proteins was evaluated using BLASTp (NCBI). Each protein sequence was queried independently against human (*Homo sapiens*; taxonomic identifier 9606) and mouse (*Mus musculus*, BALB/c strain; taxonomic identifier 10090) protein datasets using the Reference Protein (RefSeq_protein) database. Searches were performed with an expectation value (E-value) threshold of 0.05. Sequence matches exhibiting less than 35% amino acid identity with query coverage exceeding 80% were flagged as potentially non-homologous to host proteins and retained for further computational analysis. This computational analysis was performed to screen for theoretical sequence similarity rather than to infer functional cross-reactivity.

## Prediction of B-cell epitopes

Linear and conformational B-cell epitopes were predicted from the recombinant proteins using BCPRED (http://ailab-projects2.ist.psu.edu/bcpred/predict.html) and ELIPRO (http://tools.iedb.org/ellipro/), respectively. For linear epitope prediction, the amino acid sequence of each protein was submitted independently to the BCPRED server. Predictions were performed using a fixed epitope length of 20 amino acids with non-overlapping constraints. Predicted epitopes with antigenicity scores meeting the predefined high-threshold criterion (score range: 0.9–1.0) were considered as antigenic [22].

For conformational epitope prediction, the selected tertiary structural model of each protein was submitted to the ElliPro tool. Parameters were set to a minimum protrusion index score of 0.5 and a maximum distance of 6 Å. Predicted conformational epitopes and their corresponding scores were compiled in tabular form, and their spatial distribution was visualized on the modeled tertiary structures. Conformational epitopes with scores ≥ 0.5 were considered as antigenic [23]. These predictions were intended to provide theoretical insights into potential epitope regions rather than to infer immunogenic performance

### Epitope conservancy analysis

The sequence conservancy of predicted B-cell epitopes was evaluated using the Epitope Conservancy Analysis Tool provided by the Immune Epitope Database (IEDB; 2024 update). Each predicted epitope sequence was analyzed against a curated dataset of 670 HBsAg amino acid sequences retrieved from the NCBI database. An identity threshold of 80% was applied, and the percentage of HBsAg sequences containing each epitope at or above this threshold was calculated. This computational analysis was performed to assess theoretical epitope conservation across circulating HBV variants rather than to infer population-level immunogenicity.

### T helper cell epitope prediction

Putative T helper (CD4+) cell epitopes were predicted from the recombinant proteins using the IEDB Analysis Resource (http://tools.iedb.org/main/). Predictions were performed with the NetMHCIIpan 4.1 algorithm using a 15–amino acid sliding window across each protein sequence. A panel of commonly used murine MHC class II alleles (H2-IA^q, H2-IA^s, H2-IA^u, H2-IA^k, H2-IA^d, and H2-IA^b) was selected. Predicted peptides were ranked based on percentile rank and predicted binding scores. Peptides with percentile ranks ≤ 0.5 or predicted binding scores exceeding 0.5 were classified as high-affinity binders. These predictions were intended to provide theoretical insights into potential T helper epitope content rather than to infer *in vivo* immunogenicity [24].

### Evaluation of expression feasibility

The theoretical feasibility of heterologous expression of the constructs in *Escherichia coli* (*E. coli*) was evaluated using multiple computational predictors. Protein solubility was assessed using the SOLpro server (https://scratch.proteomics.ics.uci.edu/), with solubility scores greater than 0.5 interpreted as indicative of a higher likelihood of soluble expression. Aggregation propensity was analyzed using the AGGRESCAN web server (http://bioinf.uab.es/aggrescan/?utm_source=chatgpt.com) to identify short aggregation-prone regions (hot spots) and their associated metrics. The presence of signal peptides was evaluated using the SignalP (v6.0) server (https://services.healthtech.dtu.dk/services/SignalP-6.0/) to assess the likelihood of signal peptide–mediated secretion. In addition, each construct was submitted to the JCat server (https://www.jcat.de/Result.jsp) for codon optimization tailored to *E. coli* expression. These *in silico* analyses were performed to generate predictive insights into expression-related features rather than to confirm experimental expression outcomes.

### Prediction of secondary and tertiary structures

The secondary structures of the designed proteins were predicted using PSIPRED (v4.0). Tertiary structures were modeled with the I-TASSER web server (https://zhanggroup.org/I-TASSER/), generating five candidate models per protein. Each model was evaluated based on Confidence Score (C-score), Template Modeling Score (TM-score), and Root Mean Square Deviation (RMSD). The model from each protein with the highest C-score and TM-score and lowest RMSD was selected for further structural refinement. Refinement was performed using the GALAXYRefine server (https://galaxy.seoklab.org/), producing five additional candidate structures per protein. The refined structures were assessed for stereochemical quality and overall structural integrity using multiple metrics, including GDT-HA, RMSD, MolProbity score, Clash

score, poor rotamer percentage, and Ramachandran favored regions. A refined model for each protein was selected and evaluated for geometric accuracy and global quality using the PROCHECK (via the PDBsum server) (https://www.ebi.ac.uk/thornton-srv/databases/pdbsum/Generate.html)) and the ProSA (https://prosa.services.came.sbg.ac.at/prosa.php) web servers. Each validated model was independently visualized using PyMOL™ (v3.1.0) tool. All computational analyses were performed to generate predictive structural insights rather than to confirm experimentally validated conformations.

## Molecular docking

A broadly neutralizing anti-HBsAg antibody structure (PDB ID: 6VJT) was used for molecular docking analysis with the designed proteins. The refined and validated tertiary structure of each antigen and the antibody structure were independently submitted to the ClusPro 2.0 server (https://cluspro.org/login.php) for antigen–antibody docking. Docking poses (30 per antigen–antibody pair) were generated, each with a unit-less relative weighted energy score that integrates van der Waals interactions, electrostatic forces, and desolvation energy. Docked complexes were clustered according to structural similarity. Representative poses from the most populated clusters were selected for further analysis. Interacting residues at the antigen–antibody interfaces of the selected docking poses were identified and analyzed using the PDBsum server (https://www.ebi.ac.uk/thornton-srv/databases/pdbsum/Generate.html). The resulting interaction profiles were used to evaluate the theoretical binding orientation and interface characteristics of each antigen–antibody complex.

## Molecular dynamics simulations

Atomistic MD simulations were performed for the selected MeRPYS1–antibody, MeRPYS2–antibody, and MeRPYS3–antibody complexes using GROMACS 2024 with the CHARMM36m force field. Each complex was solvated in a cubic simulation box containing TIP4P water molecules, with a minimum solute–box distance of 1.0 nm, and neutralized by adding NaCl to a final concentration of 0.15 M. Energy minimization was carried out using the steepest descent algorithm until the maximum force fell below 1000 kJ·mol$^{-1}$·nm$^{-1}$.

System equilibration was performed in two stages: a 100 ps NVT equilibration at 300 K using the velocity-rescaling (V-rescale) thermostat, followed by a 100 ps NPT equilibration at 1 bar using the Parrinello–Rahman barostat. MD simulations were conducted for 110 ns with a 2 fs integration time step under periodic boundary conditions, and trajectory coordinates were saved every 100 ps.

Trajectory analyses were performed using built-in GROMACS utilities and custom Python scripts to characterize global and local structural dynamics. Structural stability was assessed by calculating the RMSD of Cα backbone atoms for each antigen–antibody complex following least-squares fitting to the initial structure. Residue-level flexibility was evaluated using root-mean-square fluctuation (RMSF) analysis of Cα atoms. Structural compactness was quantified by calculating the radius of gyration (Rg), and solvent exposure was evaluated using solvent-accessible surface area (SASA). Free-energy landscapes (FELs) were constructed based on principal component analysis of the first two eigenvectors derived from RMSD and Rg distributions, with relative free energies reported in arbitrary units.

Antigen–antibody interface stability was further examined through hydrogen-bond analysis by monitoring donor–acceptor pairs throughout the trajectories. Hydrogen-bond persistence, average donor–acceptor distances, and average and maximum continuous lifetimes were calculated. All MD simulations were conducted using fixed initial structures and predefined parameters, with stochastic elements generated using default random seeds to ensure repeatability of the simulation workflow.

## Ethics statement

This is an observable study; no new human data were generated. Ethical approval is not applicable here.

## Results and discussion

We report a computational analysis of recombinant HBsAg antigens incorporating both wild-type sequences and EMAVs. Analysis of publicly available HBsAg sequence data from Ethiopia identified regionally prevalent EMAVs within the MHR, including the "a" determinant. These observations informed the design of three recombinant constructs intended to represent the observed sequence diversity of circulating HBV strains. The constructs were evaluated *in silico* to examine predicted structural and immunological features. This workflow (Fig 1) was applied to generate testable hypotheses regarding protein design and characterization rather than to demonstrate diagnostic performance.

Key diagnostic EMAVs are defined as amino acid substitutions within the MHR of HBsAg, particularly the "a" determinant. Such variants have been associated with false-negative serological findings in previous studies. In this analysis, EMAVs classified as key include A128V, P127T, Y134F, Y134N, S143L, and S143T, based on published reports suggesting potential effects on HBV detectability rather than direct functional confirmation [8,9].

### Molecular characteristics of HBV in Ethiopia

Analysis of 670 individual patient samples with complete MHR data from Ethiopia suggested that genotype A was the predominant one in the dataset. Genotype A accounted for 488 sequences (72.84%, 95% CI: 69.5–76.2%), while genotype D represented 182 sequences (27.16%, 95% CI: 23.8–30.5%).

Among genotype A sequences, 93 contained at least one mutation within the MHR, yielding a mutation prevalence ($pA = xA/nA$) of 19.06% (95% CI: 15.6% – 22.5%). In contrast, 127 genotype D sequences harbored one or more MHR mutations (= $xD/nD$ = 69.78% (95% CI: 63.1% – 76.5%).

The pooled proportion of sequences with MHR mutations across both genotypes was calculated as:

$$P_{pooled} = \frac{xA + xD}{nA + nD} = \frac{93 + 127}{488 + 182} = \frac{220}{670} = 0.3284.$$

Using this pooled estimate, the standard error (SE) of the difference in proportions was:

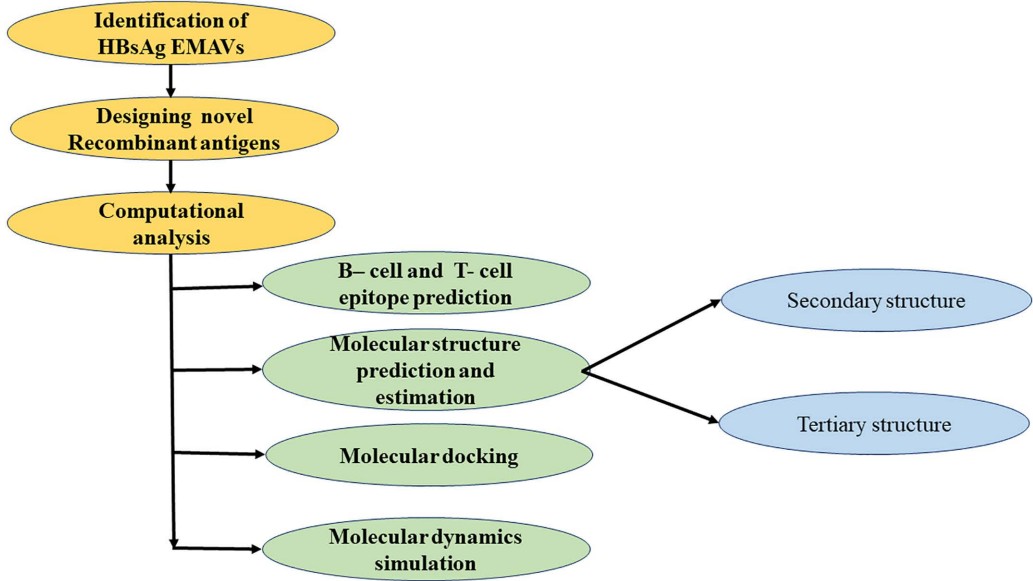

**Fig 1. Critical steps for *in silico* design and characterization of novel recombinant antigens.**

$$SE = \sqrt{Ppooled\ (1-\ Ppooled)\ (1/nA + 1/nD)}$$

$$SE = \sqrt{0.3284 \times 0.6716 \times (0.00205 + 0.00549)} = 0.0408.$$

The z-score was calculated as:

$$z = (pD - pA)/SE = (0.6978 - 0.1906)/0.0408 = 12.44.$$

This z-value corresponds to a two-tailed p-value < 0.0001, indicating a statistically significant difference in the prevalence of EMAVs between genotypes A and D.

The effect size, expressed as the absolute difference in proportions, was:

$$\triangle p = |pD - pA| = 0.5072\ (50.7\%).$$

This study demonstrates a pronounced and statistically robust difference in the prevalence of EMAVs between HBV genotypes A and D circulating in Ethiopia. Genotype D showed a substantially higher proportion of sequences carrying MHR mutations compared with genotype A, with a highly significant z-score (z = 12.44, p < 0.05).

Importantly, the observed difference was not only statistically significant but also biologically meaningful, as reflected by the large effect size (Δp = 50.7%). Reporting the effect size alongside the p-value provides essential context, indicating that the disparity in mutation prevalence is substantial and unlikely to be attributable to sampling variability alone.

Several common diagnostic escape mutations were identified, including A128V, Y134F, Y134N, S143L, and S143T. The P127T mutation was particularly frequent in genotype D, occurring in 92 cases (50.55%, 95% CI: 43.3% – 57.8%). Notably, some of these substitutions were located within the "a" determinant of both genotypes A and D (Fig 2).

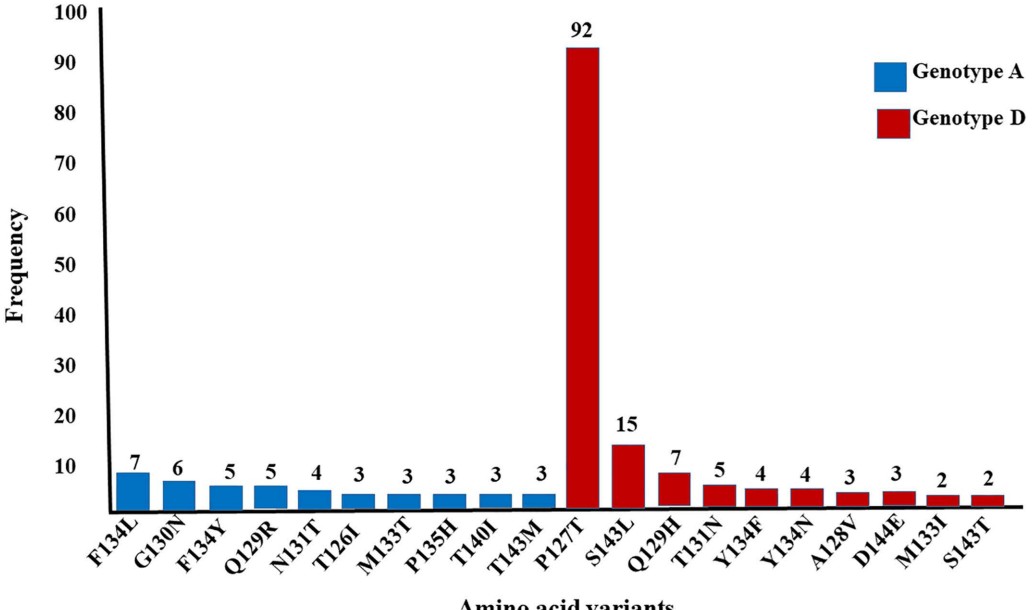

**Fig 2. The frequency of top ten most prevalent mutant amino acid variants in the "a" determinant regions of genotype A and D of HBsAg in Ethiopia.**

Our findings aligned with other previous studies. A study on pregnant women in the Amhara National Regional State of Ethiopia found that genotype A was predominant, with samples showing surface gene mutations in the MHR [25]. Another study identified a novel HBV sub genotype D10 circulating in Ethiopia, highlighting the virus genetic diversity in the country [16].

The emergence of HBsAg sequence variants could be attributed to the lack of proofreading during the reverse transcription stage of viral replication [11]. It potentially leads to mutations that can result in key diagnostic EMAVs. The phenomenon has been observed globally. For example, a study on Tunisian patients identified naturally occurring EMAVs of HBV, emphasizing the worldwide relevance of the EMAVs [9].

Recent reports from different geographic regions further support the rationale of the present study. A study from the United States identified multiple diagnostic EMAVs within the HBsAg, particularly in "a" determinant, highlighting their effects on the performance of existing diagnostic assays [26]. A study from Nigeria documented numerous EMAVs across the MHR of HBsAg, indicating their potential impact on antigenicity, diagnostic detection, and vaccine-induced immunity [27]. Liu et al. demonstrated that EMAVs can lead to false-negative results in HBV detection [28].

Collectively, these observations motivate further exploration of antigen design strategies that consider both wild-type and EMAVs of HBsAg. The computational designs presented here provide a conceptual framework for examining region-specific recombinant antigen rather than evidence of diagnostic performance. These in silico analyses are intended to generate testable hypotheses regarding sequence prioritization and construct architecture, which may inform subsequent experimental studies without presuming practical diagnostic utility.

## Computationally designed recombinant proteins

Fig 3 shows the positions of the MHR and the "a" determinant within HBsAg, along with three computationally designed recombinant proteins: MeRPYS1, MeRPYS2, and MeRPYS3. The MHR is situated near the midpoint of HBsAg and the "a" determinant is located centrally within the MHR (Fig 3a). This ensures that MeRPYS3, which is derived from the MHR, inherently includes the "a" determinant (Fig 3d).

As illustrated in Fig 3, three recombinant proteins (MeRPYS1–3) were computationally designed to explore how genotype prevalence and EMAV distribution in Ethiopia could inform region-specific antigen architectures.

MeRPYS1 (42.68 kDa; 415 amino acids) was designed as a multi-meric construct composed of ten HBsAg "a" determinant segments. The sequence begins with an N-terminal B-1-tag and ends with a C-terminal hexa-His-tag. Both wild-type and EMAV sequences from genotypes A and D were incorporated in the design. Each unit is separated by a flexible GSGSG linker to minimize steric constraints. Among the ten incorporated segments, six EMAVs derived from genotype A were distributed across four segments, while eight EMAVs from genotype D were incorporated within two segments. The remaining four segments consisted of wild-type sequences, evenly drawn from both genotypes. This composition reflects an attempt to balance the higher circulation frequency of genotype A with the relatively higher EMAV burden observed in genotype D.

MeRPYS2 (42.66 kDa; 415 amino acids) followed a more conservative design strategy. The construct contains ten "a" determinant segments linked by the GSGSG spacers, preceded by a B-1-tag and terminated with a hexa-His-tag. Most of the segments were wild-type, with six derived from genotype A and two from genotype D. Only a single EMAV, P127T from genotype D, was included and represented twice in the construct. The selective inclusion of this variant was informed by its relatively high prevalence within genotype D. The predominance of genotype A wild-type segments reflects its higher overall representation among circulating Ethiopian HBV sequences.

The contrasting designs of MeRPYS1 and MeRPYS2 were informed by statistical analysis. Genotype A accounts for more than twice the proportion of circulating HBV sequences compared with genotype D, whereas genotype D harbors a slightly higher proportion of EMAV-containing sequences. Rather than favoring a single genotype or mutation profile, the two constructs were designed to explore different weighting strategies between genotype prevalence and mutation density. This generates comparative hypotheses for subsequent experimental evaluation.

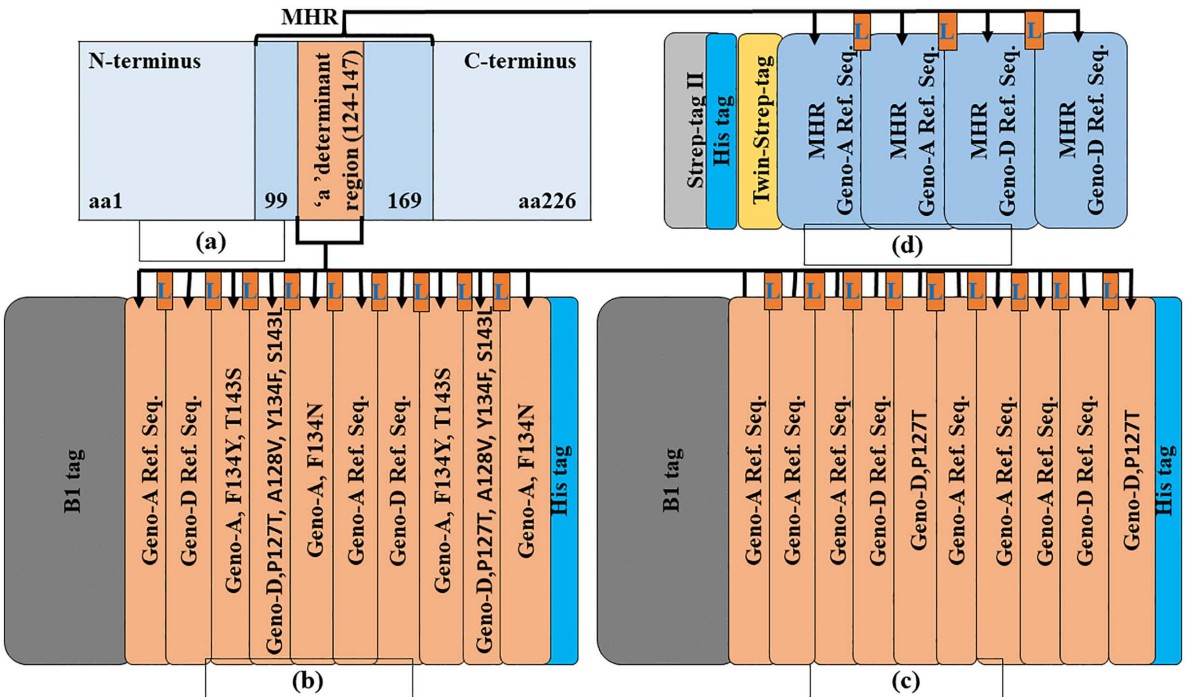

**Fig 3. Graphic illustration of computationally designed novel recombinant antigens.** (a) MHR and "a" determinant within HBsAg (b) MeRPYS1 (c) MeRPYS2 (d) MeRPYS3. L represents linker (GSGSG).

MeRPYS3 (35.67 kDa; 338 amino acids) was designed as a structurally distinct construct focusing on the MHR rather than repeated "a" determinants. It includes four wild-type MHR segments, two from genotype A and two from genotype D, separated by the GSGSG linkers. The construct contains a combination of affinity-tags (Strep-tag II, hexa-His-tag and Twin-Strep-tag) at the N-terminus. Unlike MeRPYS1 and MeRPYS2, MeRPYS3 does not incorporate EMAVs and serves as a comparative design to explore genotype-balanced, mutation-free MHR representations.

Collectively, these designs reflect a structured, data-driven exploration of how region-specific genotype distribution and EMAV prevalence may be incorporated into recombinant antigen design using in *silico* approaches. The resulting constructs are intended to support hypothesis generation regarding antigen composition and representation, rather than to assert diagnostic performance, and warrant further experimental investigation.

The inclusion of a B-1-tag at the N-terminus of MeRPYS1 and MeRPYS2 designs was intended to support exploratory expression of the recombinant proteins in *E. coli*. This tag may also enable protein detection by Western blotting using labeled anti-protein G secondary antibodies without the requirement for a primary antibody [29]. In MeRPYS3 design, multiple fusion-tags were incorporated to facilitate comparative assessment of expression and purification behavior in a bacterial system [30]. The introduction of GSGSG linker sequences segments may allow conformational freedom and reduce potential steric constraints during folding [31]. In addition, hexa-His-tags were included in all three constructs to facilitate affinity-based purification strategies, providing a theoretical framework for subsequent experimental evaluation [32].

### *In silico* characterization

**Protein homology tests.** We conducted homology tests on each computationally designed recombinant protein. The results predicted that, none of the three proteins showed significant similarity to either human or mouse proteins.

The observed lack of sequence homology suggests the possibility of increased sequence distinctiveness, which may be relevant for reducing theoretical cross-reactivity. However, this observation is based solely on computational comparison and is intended to generate hypotheses regarding specificity, which require experimental evaluation to determine their practical significance.

The significance of homology testing in developing diagnostic assays is underscored in a previous study, highlighting the challenges of highly homologous genes in molecular diagnostics. The study also emphasizes that recombinant antigens with high homology to host proteins could lead to false-positive results during diagnosis [33].

**Predicted B-cell epitopes.** The BCPRED tool predicted ten linear B-cell epitopes from each of MeRPYS1 and MeRPYS2 and eight from MeRPYS3. Each predicted epitope (20 amino acids in length) has an antigenicity score of one.

MeRPYS1 contains five unique predicted linear B-cell epitopes, with each unique epitope duplicated twice at different positions within the sequence. The predicted epitopes align with the design of MeRPYS1, which incorporates five distinct "a" determinant regions, each duplicated twice at the corresponding positions.

MeRPYS2 contains three unique predicted epitopes. The one with six duplicates at different positions of the sequence are located to wild-type "a" determinant regions of genotype A. Another one with two duplicates at different positions are located at the wild-type "a" determinant regions of genotypes D. Additionally, the third one (with two duplicates) is located within a genotype D "a" determinant region containing the P127T EMAV. These predictions are consistent with the design of MeRPYS2, which incorporates three different "a" determinant regions.

MeRPYS3 has four unique linear B-cell epitopes, two epitopes located at the MHRs of genotype A and the other two are located at the MHRs of genotype D sequences. These predictions align with the design of MeRPYS3, which includes MHR sequences from both genotypes, A and D.

Overall, the predicted distribution and characteristics of linear B-cell epitopes (Table 1) are consistent with the intended designs of the respective recombinant proteins.

Each predicted linear B-cell epitope in MeRPYS1 and MeRPYS2 corresponds to the first 19 or 20 amino acids of the respective "a" determinant. This suggests that each segment of the "a" determinant region potentially serves as a linear B-cell epitope within the antigens. This observation aligns with other studies that identified the "a" determinant as a key immunogenic and diagnostic target for HBV [34].

Each of the five distinct B-cell epitopes in MeRPYS1 may be recognized by at least one distinct antibody. Consequently, the entire antigen may support polyclonal antibody responses during animal inoculation [7]. The same logic may work for MeRPYS2 and MeRPYS3 antigens.

**Table 1. Linear B-cell epitopes predicted from computationally designed novel recombinant antigens.**

| MeRPYS1 (aa1–415) | | MeRPYS2 (aa1–415) | | MeRPYS3 (aa1–338) | |
|---|---|---|---|---|---|
| **Epitope** | **Position** | **Epitope** | **Position** | **Epitope** | **Position** |
| **CTTPAQGNSMFPSCCCTKPT** | aa125–144 | CTTPAQGNSMFPSCCCTKPT | aa125–144 | PGSTTTSTGPCKTCTTPAQG | aa52–71 |
| **CTTPAQGTSMYPSCCCTKPS** | aa154–173 | CTTPAQGNSMFPSCCCTKPT | aa154–173 | FPSCCCTKPTDGNCTCIPIP | aa75–94 |
| **CTTPAQGNSMYPSCCCTKPS** | aa183–202 | CTTPAQGNSMFPSCCCTKPT | aa183–202 | PGSTTTSTGPCKTCTTPAQG | aa128–147 |
| **GCTTTVQGTSMFPSCCCTKP** | aa211–230 | CTTPAQGTSMYPSCCCTKPS | aa212–231 | FPSCCCTKPTDGNCTCIPIP | aa151–170 |
| **CTTPAQGNSMNPSCCCTKPT** | aa241–260 | CTTTAQGTSMYPSCCCTKPS | aa241–260 | PGSSTTSTGPCRTCTTPAQG | aa204–223 |
| **CTTPAQGNSMFPSCCCTKPT** | aa270–289 | CTTPAQGNSMFPSCCCTKPT | aa270–289 | PSCCCTKPSDGNCTCIPIPS | aa228–247 |
| **CTTPAQGTSMYPSCCCTKPS** | aa299–318 | CTTPAQGNSMFPSCCCTKPT | aa299–318 | PGSSTTSTGPCRTCTTPAQG | aa280–299 |
| **CTTPAQGNSMYPSCCCTKPS** | aa328–347 | CTTPAQGNSMFPSCCCTKPT | aa328–347 | PSCCCTKPSDGNCTCIPIPS | aa304–323 |
| **GCTTTVQGTSMFPSCCCTKP** | aa356–375 | CTTPAQGTSMYPSCCCTKPS | aa357–376 | | |
| **CTTPAQGNSMNPSCCCTKPT** | aa386–405 | CTTTAQGTSMYPSCCCTKPS | aa386–405 | | |

However, it is important to note that the predicted epitopes have not yet been validated experimentally. Antigenicity predictions do not always reflect actual antibody recognition because epitope exposure can be influenced by protein folding, conformational dynamics, and post-translational modifications in the eukaryotic expression system. Therefore, experimental validation is required before concluding that the predicted regions are truly antigenic. This limitation is consistent with the findings in a previous study, reporting that the sequence-based epitope prediction offers only modest discrimination between epitope and non-epitope residues. Accordingly, these predictions should be used primarily to prioritize or filter candidate antigens for subsequent experimental validation [35].

*In silico* prediction of conformational epitopes using ELIPRO identified three to five discontinuous epitopes in each recombinant protein (Table 2). In MeRPYS1, the highest scoring conformational epitope (aa363–413, score 0.844) coincided with the C-terminal cluster of linear epitopes predicted by BCPRED. Similarly, in MeRPYS2, the largest conformational patch (aa320–413, score 0.764) overlapped multiple linear epitopes in the mid-to-C-terminal region. For MeRPYS3, ELIPRO predicted multiple conformational epitopes (aa269–271, aa274; aa141–151, aa158–170, aa187–194, aa229–245, aa324, aa334) that corresponded closely with BCPRED predicted linear epitopes.

A strong concordance was observed between linear and conformational epitope predictions across all three proteins, with most linear epitopes either fully encompassed by or overlapping ELIPRO-predicted conformational patches (Fig 4). This alignment provides predictive insight into surface-exposed antigenic regions.

The incorporation of repeated similar epitopes reflects a speculative computational design approach and does not constitute confirmatory evidence of enhanced epitope presentation or immune recognition [36]. Epitope duplication does not inherently confer immuno-dominance, which is a multifactorial phenomenon influenced by MHC binding affinity, antigen processing efficiency, epitope competition, T-cell repertoire, and host-specific immune regulation [37].

## Epitope conservancy analysis result

The IEDB Epitope Conservancy Tool evaluated the conservancy of the predicted B-cell epitopes across 670 HBsAg sequences at 80% identity threshold. The result showed that the majority of epitopes were conserved, with most achieving >95% sequence coverage at or above the threshold. Among the five epitopes predicted from MeRPYS1, four exhibited >95% coverage, while the remaining one displayed 94.4% coverage. All predicted epitopes from MeRPYS2 and MeRPYS3 demonstrated near-universal conservation (>98% coverage). The partial variability observed in MeRPYS1 epitopes attributed potentially to the inclusion of EMAVs in protein design. Overall, these findings indicate that the predicted epitopes are conserved across diverse HBsAg amino acid sequences.

**Table 2. Predicted conformational B-cell epitopes, their positions within the respective protein sequence, number of residues, and their antigenicity scores.**

| | MeRPYS1 | | | MeRPYS2 | | | MeRPYS3 | | |
|---|---|---|---|---|---|---|---|---|---|
| No. | Residues | | Score | Residues | | Score | Residues | | Score |
| | Position | # | | Position | # | | Position | # | |
| 1 | 363-413 | 51 | 0.844 | 320-413 | 93 | 0.764 | 269-271, 274 | 4 | 0.785 |
| 2 | 182, 193-228 | 37 | 0.670 | 237, 242, 253-293 | 43 | 0.628 | 141-151, 158-170, 187-194, 229-245, 324, 334 | 51 | |
| 3 | 247-258 | 12 | 0.614 | 156-164, 171-177 | 16 | 0.540 | 1-13, 15–20, 22–23, 25–59, A:G60, 64–74, 81–88, 113–138, 204, 207–215, 249, 252–257, 260–268 | 126 | 0.663 |
| 4 | 296-316 | 21 | 0.599 | | | | 154, 182-86, 331, 333, 335-338 | 10 | 0.615 |
| 5 | 355-362 | 8 | 0.566 | | | | | | |

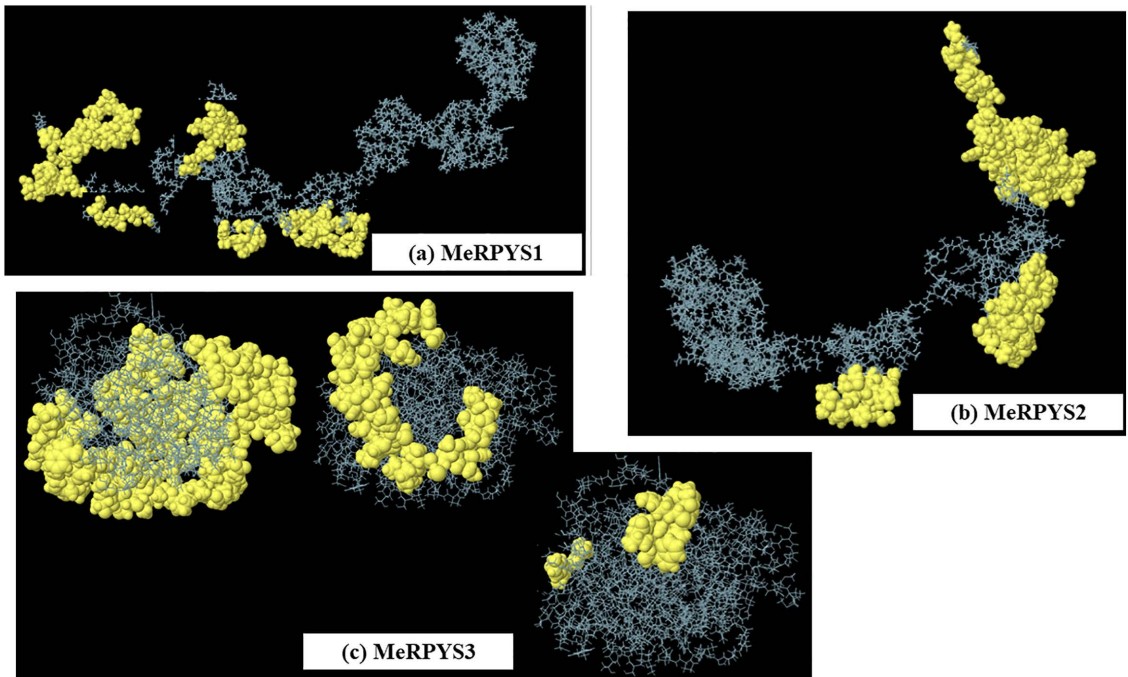

**Fig 4. Spatial distributions of conformational epitopes predicted by ELIPRO.** The areas colored yellow represent the conformational B-cell epitopes of each antigen.

## Predicted T- helper cell epitopes

IEDB predicted potential T-helper epitopes from recombinant proteins. MHC class II binding predictions using common murine alleles resulted in a dominant set of epitopes with stable interaction patterns. Notably, the peptide SPSLNAAK-SELAEAK (core: LNAAKSELA) was a potential binder across multiple alleles, exhibiting percentile ranks as low as 0.01. Overlapping epitopes, including KSPSLNAAKSELAEA and PSLNAAKSELAEAKK, were predicted to be potential binders to several alleles.

The conservation of these epitopes among the three recombinant proteins suggests that they may function as CD4$^+$T-cell targets, potentially facilitating B-cell activation and promoting the production of cross-reactive antibodies. However, the predicted T-cell epitopes have not yet been experimentally validated. These predictions are probabilistic rather than definitive, and therefore require experimental confirmation, as emphasized in previous study by Fleri *et al* [38].

## Expression feasibility

The SOLpro solubility prediction indicated that all the proteins are likely to be expressed in *E. coli,* with probable solubility scores of 0.813 (MeRPYS1), 0.839 (MeRPYS2), and 0.980 (MeRPYS3). Scores above the threshold (0.5) suggested a high likelihood of soluble protein expression in *E. coli.*

The Aggrescan aggregation prediction revealed that there is low aggregation propensity for all proteins. MeRPYS1 contains six hot spots (HS) with a normalized aggregation score (Na4vSS) of −20.1. MeRPYS2 has four HS and a Na4vSS of −20.2. MeRPYS3 also contains four HS and displayed the highest area above threshold (AAT = 34.128), but the Na4vSS (−4.1) remained negative. Overall, the outputs from these analyses suggest that the proteins are predicted to be largely non-aggregation-prone, with only MeRPYS3 showing a slightly higher aggregation tendency.

The SignalP (v6.0) server predicted no signal peptides from all the antigens, suggesting that none of the proteins is predicted to be directed through the secretory pathway.

Each protein sequence was further subjected to codon optimization using the JCat server for expression in *E. coli*. The codon optimized sequences demonstrated a CAI value of one for all the proteins, reflecting potential compatibility with the *E. coli* codon usage. The GC contents (54.6% for MeRPYS1, 54.8% for MeRPYS2 and 57.9% for MeRPYS3) were within the optimal range for efficient gene transcription and translation in *E. coli* [39].

## Predicted secondary structures

The PSIPRED tool was used to predict the secondary structure of each designed protein, revealing proportions of alpha helices, extended strands, and random coils (Fig 5a–5c). These predictions highlight computational trends in the distribution of secondary structural elements and suggest regions of potential rigidity and flexibility within the modeled sequences. While these observations provide insights into structural patterns that may influence protein behavior, they are predictive in nature and do not confirm functional conformation in a biological environment [40].

The proportions of helix, strand, and coil in the secondary structure of MeRPYS1 were 15%, 7%, and 78%, respectively. In MeRPYS2, these values were 15%, 5%, and 80%; while in MeRPYS3, they were 15%, 15%, and 70%, respectively.

Our findings align with a study published on designing multi-epitope-based vaccine targeting M-protein of SARS-CoV2, where PSIPRED predicted the secondary structure of the construct. The study reported a proportion of helices (39%),

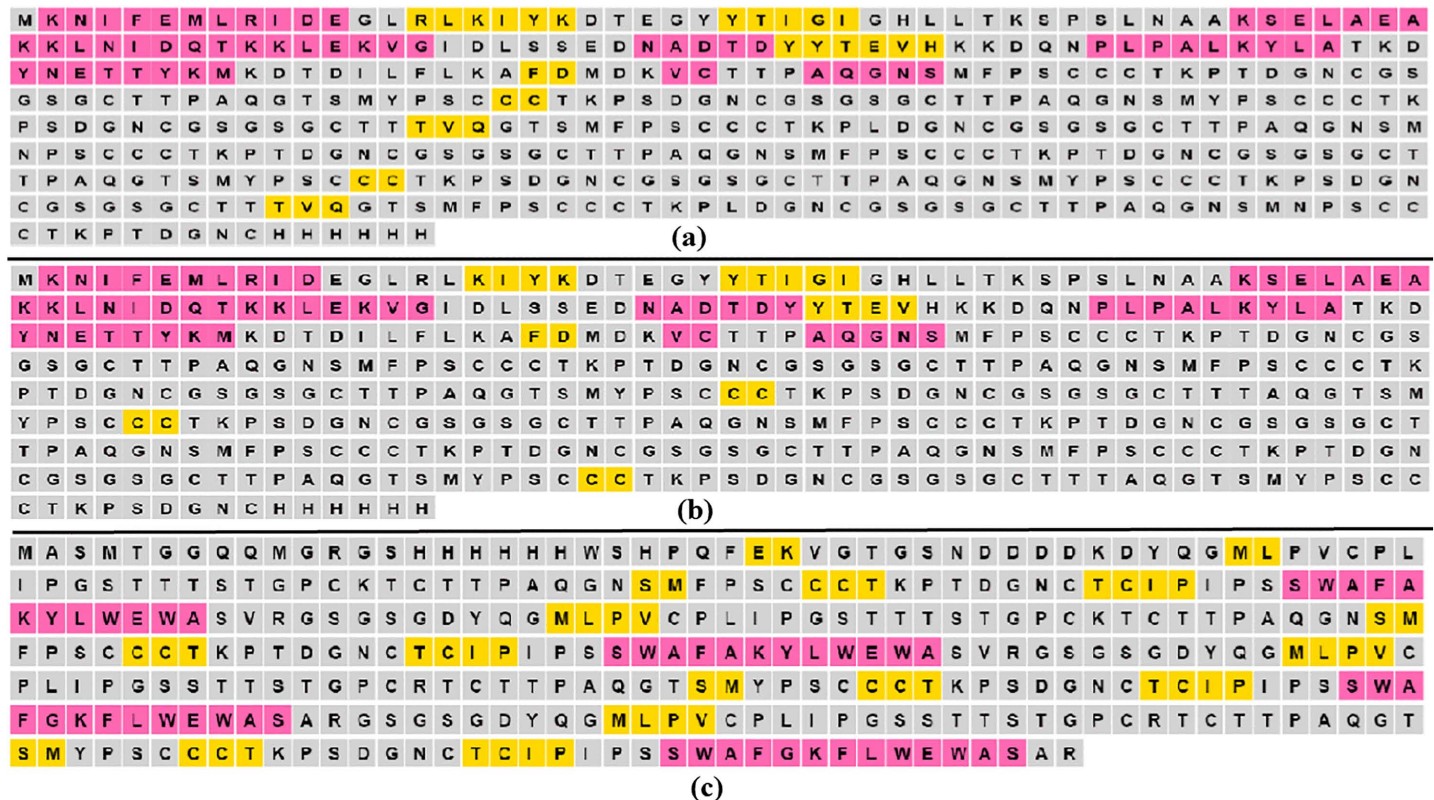

**Fig 5. Secondary structures of computationally designed antigens generated by PSIPRED tool.** (a) MeRPYS1 (b) MeRPYS2 (c) MeRPYS3. The areas shaded Pink, Yellow and Gray, represent helix, strand and coil, respectively.

strands (16%) and coils (44%) [41]. Similarly, PSIPRED was employed to predict secondary structure in the design of a multi-epitope vaccine against SARS-CoV-2, emphasizing the significance of structural analysis [42].

### Predicted tertiary structures

`The I-TASSER web server generated five PDB-format tertiary structures for each computationally designed protein. For MeRPYS1, the model with the highest C-score (−1.26) and TM-score (0.56±0.15), along with the lowest RMSD (9.8±4.6 Å), was selected for further analysis. The C-score and TM-score of the selected model suggest a globally plausible topology, while high RMSD value indicates potential deviations in local regions. Overall, these metrics indicate that I-TASSER identified structurally related but distant templates. This provides computational support for the overall fold while highlighting regions that may have been modeled with lower confidence. These observations are intended to guide hypotheses for subsequent structural and experimental analyses, rather than confirming functional or structural accuracy.

I-TASSER threading analysis identified PDB entry ID: 8wxbY as the closest structural analog for MeRPYS1, with a TM-score of 0.832 and RMSD of 2.93 Å, covering approximately 90% of the sequence. Although the overall sequence identity was low (~10%), the relatively high TM-score suggests that the global fold was reasonably captured by the modeling approach. The remaining ~10% of the sequence, mainly loop and terminal regions, lacked strong template coverage and was therefore modeled *ab initio*. These regions likely contributed to the higher global RMSD (9.8±4.6 Å) and should be considered lower-confidence areas within the predicted model. Similarly, for MeRPYS2, the model with the highest C-score (−1.46) was selected, providing a computationally guided representation of the possible global fold, which can be used to generate hypotheses for further structural and experimental validation.

For MeRPYS3, the model with the highest C-score (−2.56) and TM-score (0.42±0.14), along with the lowest RMSD (12.5±4.3 Å), was selected for further analysis. Template analysis indicated structural similarity with several experimentally validated proteins. The top five ranked analogs (PDB: 6mk2A, 4ke4A, 4g0bA, 6lpvA, and 6dd2A) displayed TM-scores greater than 0.87, RMSD values between 1.0–3.0 Å, and template coverage exceeding 95%. These metrics provide computational support for the plausibility of the overall fold and suggest potential alignment with known structural patterns. Core secondary-structure elements showed higher confidence and alignment with the template structures, while certain surface-exposed loops and terminal regions were modeled with lower accuracy, reflecting areas of lower confidence. Overall, these observations highlight predictive structural trends that can guide hypotheses for further computational and experimental validation.

All of the selected models were visualized by PyMOL software (Fig 6a-6c), showing the fusion tag, the linker peptides and protein domains within the tertiary structures.

The GALAXY Refine server generated five refined structures per protein from the selected I-TASSER predicted model. The refined structures vary with GDT-HA, RMSD, MolProbity, Clash score, Poor rotamers and Ramachandran favored region. The model with the highest GDT-HA value, was selected for each protein (Table 3).

The refined models were visualized by PyMOL software (Fig 6d–6f). Evaluation of the refined models using ProSA provided Z-scores (Table 3) and generated images illustrate the overall structural quality of each model (Fig 7a–7c).

PROCHECK via the PDBsum web server evaluated the stereo-chemical accuracy of each selected structure and then generated corresponding Ramachandran plots (Fig 8a–8c). The Ramachandran plots allowed us to determine the predicted distribution of residues within the favored, allowed and disallowed regions, providing further validation of the structural quality of each protein.

As shown in Table 4, the majority of residues (>95%) were located within the favored and allowed regions of the respective Ramachandran plots, with fewer than 5% in disallowed regions. These results suggest that the modeled tertiary structures of the proteins exhibit acceptable stereo-chemical quality under the applied computational assessment, providing predictive information for further *in silico* analyses and experimental validations. This observation is consistent with trends reported in previous studies on multi-epitope vaccine design and immune-molecular analyses of divergent human

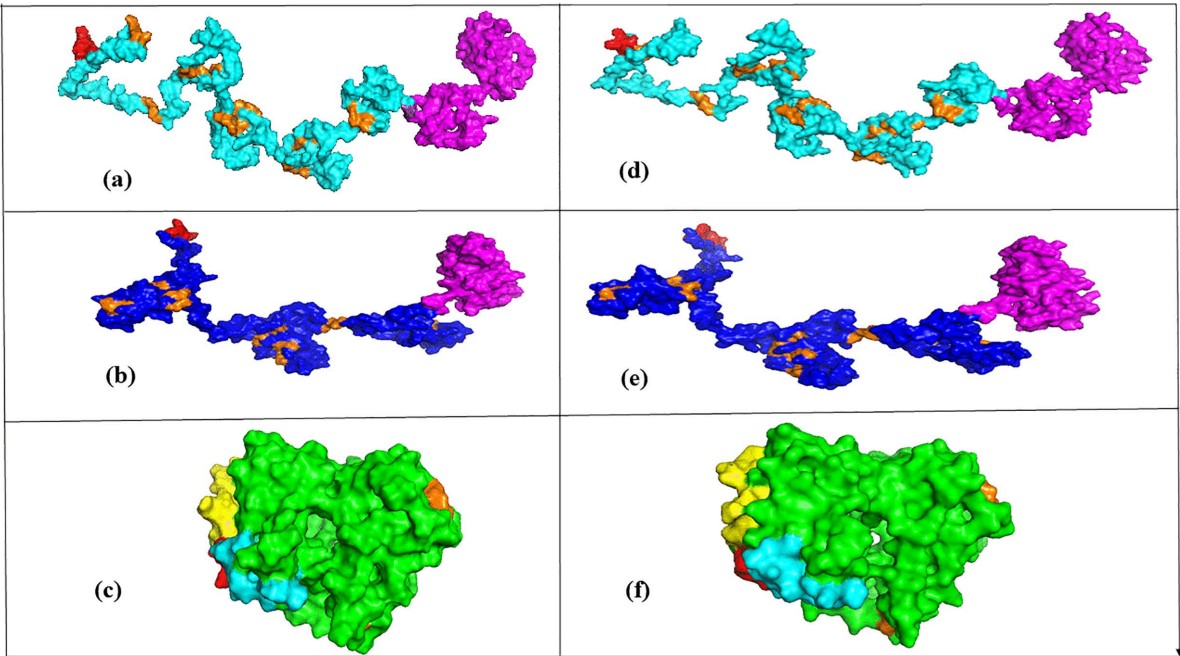

**Fig 6. Tertiary structures predicted by I-TASSER and refined by GALAXY web servers.** Tertiary structures: (a) MeRPYS1 (b) MeRPYS2 (c) MeR-PYS3; Refined structures: (d) MeRPYS1 (e) MeRPYS2 (f) MeRPYS3; protein domains: light blue, deep blue, green; B-1-tags: magenta; flexible linkers: orange; His-tags: red; and Strep-tag II: yellow.

**Table 3. The values of selected parameters for tertiary structures after GALAXY refinement and ProSA evaluation.**

| Recombinant protein | Selected Model | GDT-HA | RMSD | MolProbity | Clash score | Poor rotamers | Rama favored | Z-Score |
|---|---|---|---|---|---|---|---|---|
| MeRPYS1 | Model 2 | 0.919 | 0.503 | 1.982 | 6.300 | 0.000 | 86.200 | −1.290 |
| MeRPYS2 | Model 2 | 0.922 | 0.507 | 2.528 | 19.800 | 0.600 | 79.900 | −2.090 |
| MeRPYS3 | Model 1 | 0.934 | 0.475 | 2.331 | 16.700 | 1.100 | 88.400 | −3.850 |

papillomavirus types. This could serve to generate hypotheses regarding the structural plausibility of the designed proteins rather than confirming their functional suitability [40].

Other studies have also highlighted the effective use of I-TASSER for structure prediction combined with Ramach-andran plot analysis for structural validation. For example, a study on the LuxI protein utilized I-TASSER for predicting tertiary structures, followed by validation using Ramachandran plots. The analysis revealed that over 90% of the residues were located in favored regions [43].

## Molecular Docking

Among the 30 molecular docking results generated by ClusPro for each protein, the model with the highest population coverage and lowest relative docking score was selected for further analysis. These selected models illustrate predicted antigen–antibody interaction patterns and provide predictive computational insights into possible binding orientations. Details of the selected models, including cluster members, cluster centers, and unit less relative docking scores, are presented in Table 5. It is important to note that these docking scores do not represent absolute binding energies, but rather serve as relative metrics for comparing predicted binding orientations across models. The results are intended to generate hypotheses about potential interactions, which require experimental validation to determine biological relevance.

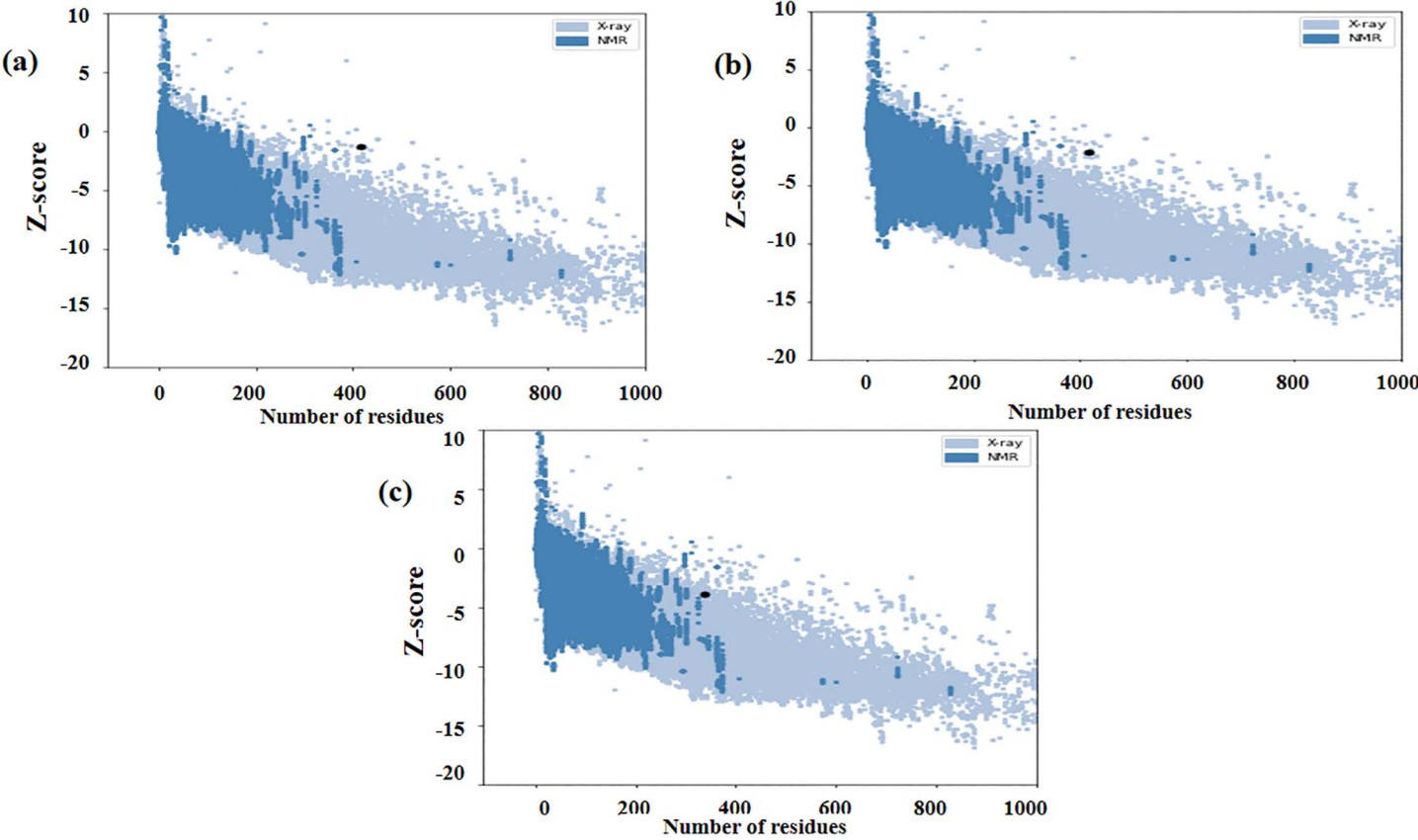

**Fig 7. Overall structural quality assessment of proteins using ProSA-web server.** (a) MeRPYS1, (b) MeRPYS2, and (c) MeRPYS3.

A previous study demonstrated that lower relative free energy values are indicative of more stable interaction patterns compared with binding poses exhibiting higher relative free energy values [44], supporting our findings.

The three selected PDB files (one for each protein) from ClusPro generated models were visualized using PyMOL, highlighting key interactions as shown in Fig 9 below.

## Molecular dynamics simulations

MD simulations performed using GROMACS were used to examine the time-dependent behavior of atoms and molecules within the MeRPYS1–3–anti-HBs modeled complexes. These simulations provided computational insights into the theoretical structural dynamics and interaction patterns of the models under simulated conditions. Analysis of the resulting trajectories (Fig 10), including RMSD, RMSF, Rg, SASA, and FEL metrics, suggests potential trends in structural behavior and interactions, which may guide further experimental studies rather than demonstrating functional binding or stability.

As shown in Fig 10, RMSD values for the modeled MeRPYS1–3–anti-HBs complexes remained below 1.75 Å over the 110 ns production trajectories. The values reached a plateau shortly after the equilibration phase. The relatively low RMSD values indicate limited deviation of the backbone-defined global conformations within the simulation framework. As RMSD was calculated using backbone atoms only, these values do not reflect the behavior of flexible regions nor imply structural rigidity. Instead, the observations suggest preservation of the docked conformations under the applied simulation conditions, providing computationally derived trends that may inform hypotheses regarding possible interaction

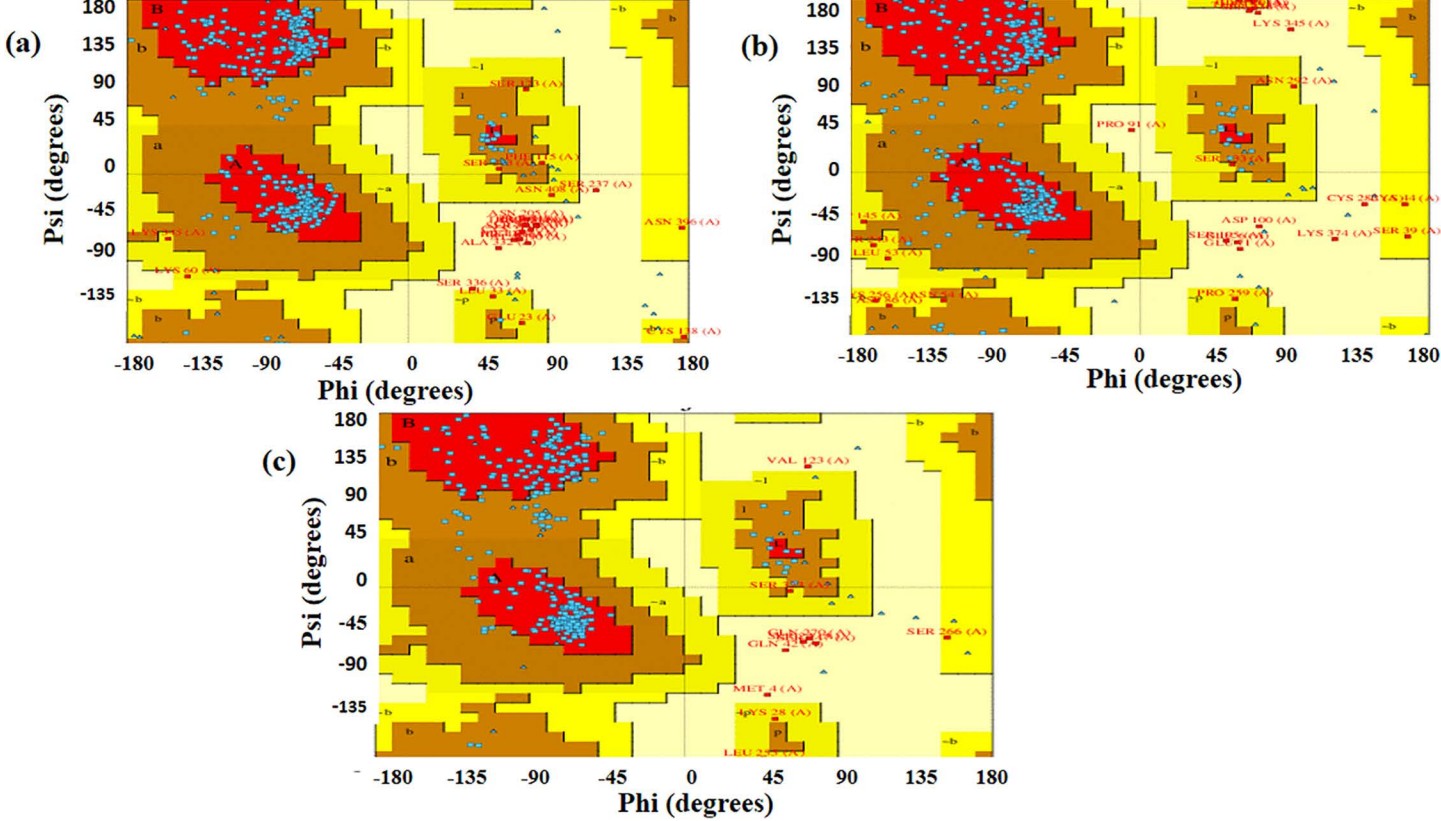

**Fig 8. Ramachandran plots.** (a) MeRPYS1, (b) MeRPYS2, and (c) MeRPYS3. Red areas (most favored regions); brown areas (additionally allowed regions); yellow areas (generously allowed regions); white areas (disallowed regions); and blue dots (amino acid residues).

**Table 4. Ramachandran plot validation results.**

|  | MeRPYS1 | MeRPYS2 | MeRPYS3 |
|---|---|---|---|
| **Most Favored Regions** | 78.800% | 71.200% | 79.000% |
| **Additional Allowed Regions** | 13.900% | 21.500% | 17.200% |
| **Generously Allowed Regions** | 3.000% | 3.900% | 1.500% |
| **Disallowed Regions** | 4.200% | 3.300% | 2.300% |

**Table 5. Relative ranking scores for the selected docked models.**

|  | Weighted Score | | |
|---|---|---|---|
|  | MeRPYS1-anti HBs | MeRPYS2-anti HBs | MeRPYS3-anti HBs |
| **Selected Cluster** | 13th | 1st | 1st |
| **Members with in the Cluster** | 16 | 69 | 134 |
| **Center (Å)** | −1205.100 | −916.400 | −919.300 |
| **Lowest Energy** | −1205.100 | −1146.700 | −1112.100 |

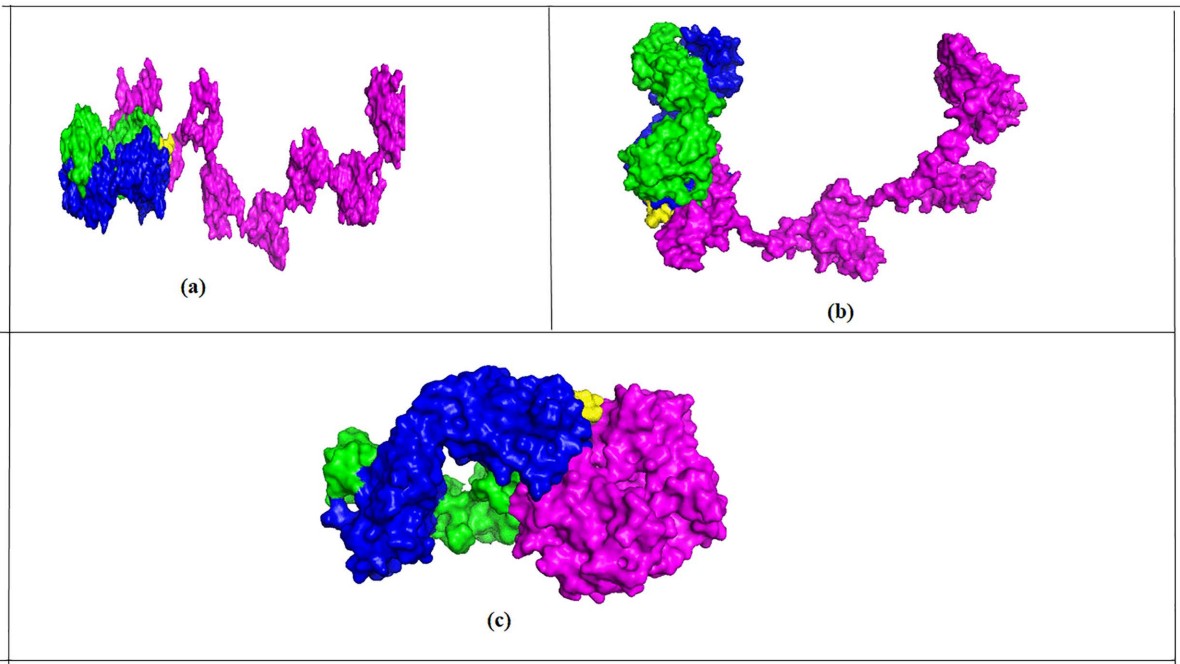

**Fig 9. Molecular docking results of each recombinant protein and anti-HBS antibody complex.** (a) MeRPYS1-anti-HBs (b) MeRPYS2-anti-HBs (c) MeRPYS3-anti-HBs. Magenta regions (recombinant proteins); green regions (heavy chains of anti-HBs antibody); and blue regions (light chains of anti-HBs antibody).

behaviors. These results do not constitute evidence of functional stability or binding and require experimental validation to assess their biological relevance.

Per-residue RMSF analysis revealed an average predicted fluctuation of approximately 0.75 Å, reflecting modest thermal motion consistent with folded globular proteins. Observed RMSF peaks were primarily located in terminal regions and solvent-exposed loops, which are inherently flexible. In contrast, core secondary-structure elements and residues involved in predicted epitope contacts showed lower fluctuations, suggesting relative structural stability in these regions under the simulated conditions. These observations highlight computational trends in the modeled complexes and may inform hypotheses regarding potential packing and interaction patterns at the antigen–antibody interface. This requires experimental validation to assess their biological significance.

The Rg values oscillated between approximately 20 and 22 Å throughout the simulation, without sustained upward drift, reflecting typical "breathing" motions of the complexes while maintaining overall compactness. These Rg patterns indicate that the global fold of the antigens was largely preserved under simulated conditions, providing computational insights into their structural behavior.

Consistently, SASA values fluctuated between 120 and 140 nm², paralleling the Rg dynamics and suggesting minor, reversible surface rearrangements. These fluctuations may correspond to transient motions in flexible loops. Together with Rg observations, these results highlight trends in simulated structural dynamics and can be used to generate hypotheses regarding the potential folding behavior and surface properties of the modeled complexes.

Finally, the FELs, projected onto the first two principal components (RMSD and Rg), displayed a single dominant basin with relatively shallow barriers (<1.0 a.u.) for all modeled complexes. Relative free-energy values ranged from 0 to 0.96 a.u. and were derived from probability density distributions of conformational states, intended for comparative visualization of conformational trends rather than absolute energetic quantification. These observations suggest that the sampled

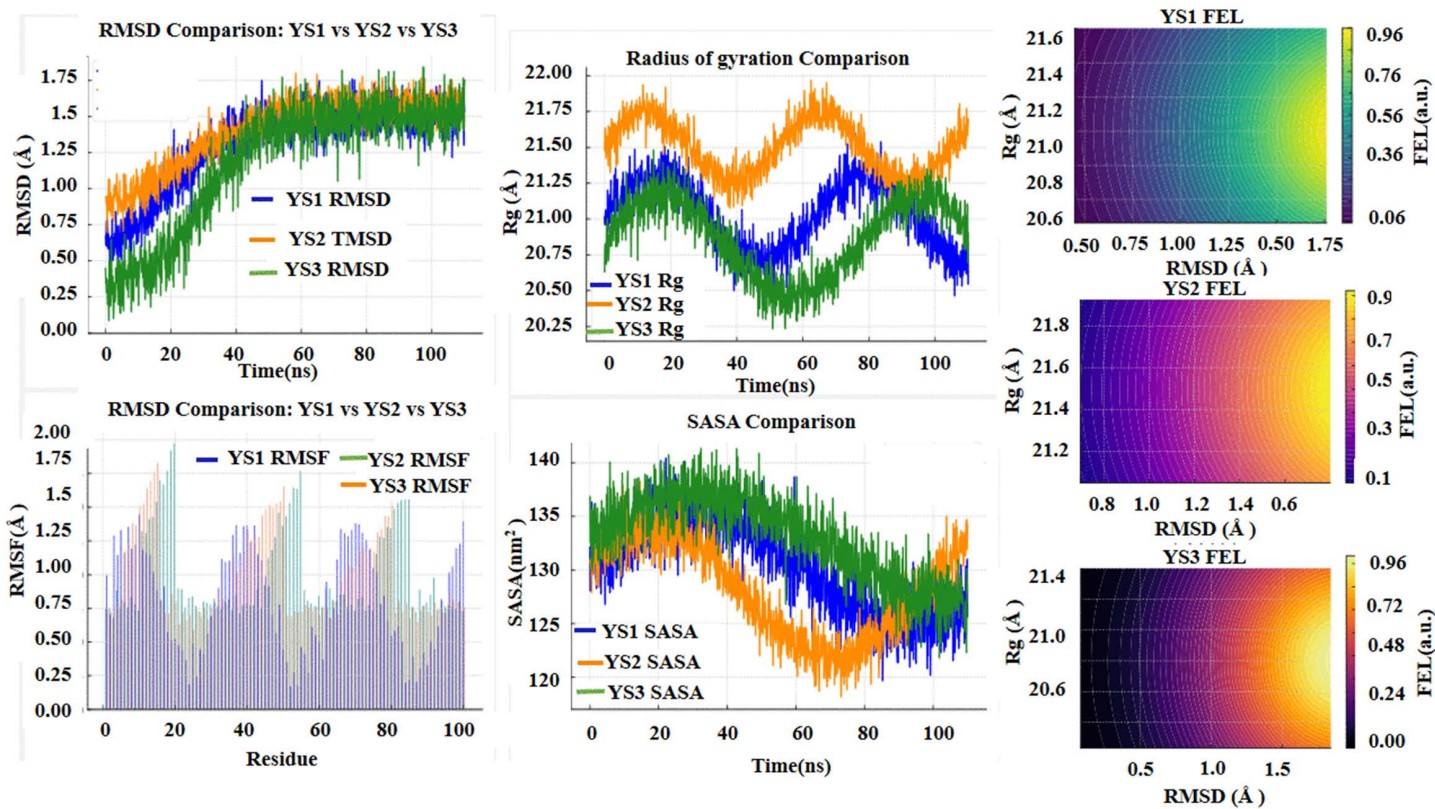

**Fig 10. MD Simulations analysis of MeRPYS1–3–anti-HBs complexes.**

conformations are closely related within the simulation framework and separated by low energetic barriers, highlighting computational trends in macromolecular behavior. The smooth landscape may inform hypotheses regarding possible transitions between energetically similar sub-states, without implying confirmed structural stability or functional patterns in biological systems.

In the MeRPYS1–anti-HBs complex, Glu45:NH and Arg102:NH were observed to form hydrogen bonds with occupancies of approximately 87.9% and 76.4%, respectively, and average lifetimes of ~11–14 ns under the simulation conditions. In MeRPYS2, hydrogen bonds exhibited lower persistence, with occupancies ranging from 40–60% and lifetimes shorter than 10 ns. In MeRPYS3, Glu45:NH and Arg102:NH showed occupancies greater than 87%, with average lifetimes exceeding 16 ns and a maximum continuous lifetime of over 40 ns. Additionally, Tyr33:OH displayed a hydrogen-bond occupancy of approximately 75%. These observations highlight computational trends in hydrogen-bond formation and occupancy within the modeled complexes. They may be used to generate hypotheses regarding potential interactions and structural behavior at the antigen–antibody interface, while experimental validation is required to assess their biological relevance.

The convergence of all evaluated parameters (RMSD, RMSF, Rg, SASA, and FEL) over the 110 ns trajectories suggests that the simulation length was sufficient to observe consistent trends in the modeled complexes. Collectively, these analyses highlight computationally derived patterns in structural behavior and dynamics. Observed features, including relatively low RMSD and RMSF values, stable Rg and SASA profiles, and shallow, single-basin FELs, indicate small-amplitude motions consistent with possible thermal fluctuations under the simulated conditions. These trends are further

complemented by hydrogen-bond analysis, which may identify potential interfacial contacts that may guide hypotheses regarding interactions at the antigen–antibody interface.

Recent studies have employed similar analyses to evaluate the stability and rigidity of vaccine candidates. For instance, a study on a novel dual-pathogen multi-epitope mRNA vaccine carried out MD simulations to assess the stability of the vaccine-TLR4 complex. The findings indicated minimal distortion in the residues of the complex showing less deviation, which supports the stability of the interaction between the mRNA vaccine and TLR4 [45]. Another study on the SARS-CoV-2 main protease and inhibitors complexes utilized MD simulations, providing insights into the stiffness and stability of the protein-ligand complexes [46].

A study on the polymerase protein for multi-epitope vaccine prediction against HBV also emphasized the importance of protein stability in vaccine design. It underscores that the stability of the DNA polymerase protein is crucial for inducing immune responses and developing effective vaccines [47]. Another study that employed MD simulations on azidolysozyme analyzed cross-correlated motions to understand the dynamic behavior of the protein. The study highlighted the significance of correlated and anti-correlated motions in protein function and stability [48].

The findings of these studies are consistent with the computational trends observed in our study. Antigen–antibody complexes analyzed in MD simulations, including our designs, exhibited modeled patterns of structural behavior and interaction tendencies. These observations may inform hypotheses regarding potential structural organization and interaction dynamics, but they do not provide evidence of confirmed rigidity, stability, or functional activity, which would require experimental validation.

## Conclusion

This study aimed to identify EMAVs circulating in Ethiopia and incorporate them into the design of region-specific recombinant proteins to guide future diagnostic development. Using *in silico* approaches, prevalent EMAVs were identified, and three recombinant proteins were designed incorporating both wild-type and EMAV sequences. Predicted B-cell and T helper cell epitopes were mapped within the designed proteins, and computational assessments indicated that the proteins could potentially be expressed in *E. coli* and purified in soluble forms. Structural modeling suggested that the antigens may adopt coherent secondary and tertiary conformations, while molecular docking and subsequent MD simulations highlighted predicted trends in potential interactions between the designed proteins and the anti-HBs antibody. Collectively, these findings generate testable hypotheses for further experimental investigation, including protein expression and purification, antibody production in animal models, and evaluation in diagnostic assays, without implying confirmed functional performance.

### Limitation of the study

We analyzed publicly available HBsAg amino acid sequence data from Ethiopian HBV isolates retrieved from the NCBI database; no new sampling or sequencing was performed. Since, the dataset depends on the scope of local surveillance and reporting, it may carry geographic sampling bias and potentially over-represent some regions of the country.

Some limitations exist within the molecular characterization of the computationally designed recombinant proteins. T-cell epitope modeling was restricted to murine MHC class II alleles, and extrapolation to human immune responses will require validation using human HLA molecules.

Another key limitation of this study is that molecular docking and MD simulations were performed using a single anti-HBs antibody structure with only one simulation per antigen–antibody complex and without replica runs. Binding free energy calculations were not performed. Consequently, the analyses provide qualitative, predictive insights into potential antigen–antibody interactions rather than quantitative measures of binding affinity or immune-reactivity.

This study employed an integrated computational workflow to design and evaluate recombinant protein constructs using multiple *in silico* tools. The findings are entirely prediction-based and should be interpreted as

hypothesis-generating. The proposed antigens have not yet undergone experimental validation, including protein expression, purification, structural characterization, or immunological assessment. Therefore, further experimental studies are required to evaluate these constructs. Such studies could include testing in specific diagnostic formats, such as ELISA or lateral flow assays, to assess their practical utility and performance relative to established reagents.

## Future directions

In our future work, the designed constructs will be expressed in *E. coli* and purified by Ni-NTA affinity chromatography. Their antigenicity will be evaluated by ELISA using panels of Ethiopian HBV-positive sera and HBV-negative controls. In addition, the purified antigens will be used to immunize Swiss albino mice for the production of polyclonal antibodies. Immunological diagnostic assays will be developed and optimized and their clinical performance will be compared with marketed HBV tests.

## Supporting information

**S1 File. Frequencies of EMAVs in the MHR of HBV genotypes A and D circulating in Ethiopia.**
(PDF)

**S1 Fig. Molecular Docking Plots of MeRPYS1–3-anti-HBs complexes, showing predicted binding residues of the antigen and antibody.**
(TIF)

## Author contributions

**Conceptualization:** Yeshwas Abite Workneh, Abraham Tesfaye Bika.

**Data curation:** Yeshwas Abite Workneh, Desye Melese Sisay.

**Formal analysis:** Yeshwas Abite Workneh.

**Funding acquisition:** Abebaw Fekadu, Abraham Tesfaye Bika.

**Investigation:** Yeshwas Abite Workneh.

**Methodology:** Yeshwas Abite Workneh, Desye Melese Sisay, Tesfaye Sisay Tessema.

**Project administration:** Abebaw Fekadu, Abraham Tesfaye Bika, Alemu Tekewe Mogus.

**Resources:** Abebaw Fekadu.

**Supervision:** Abraham Tesfaye Bika, Alemu Tekewe Mogus, Tesfaye Sisay Tessema.

**Validation:** Yeshwas Abite Workneh, Desye Melese Sisay.

**Visualization:** Yeshwas Abite Workneh, Desye Melese Sisay.

**Writing – original draft:** Yeshwas Abite Workneh.

**Writing – review & editing:** Yeshwas Abite Workneh, Abraham Tesfaye Bika, Alemu Tekewe Mogus, Tesfaye Sisay Tessema.

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
