## [Decision Letter · Decision Letter 0]

20 Aug 2025

Dear Dr. Workneh,

Thank you for submitting your manuscript to PLOS ONE. After careful consideration, we feel that it has merit but does not fully meet PLOS ONE’s publication criteria as it currently stands. Therefore, we invite you to submit a revised version of the manuscript that addresses the points raised during the review process.

We look forward to receiving your revised manuscript.

Kind regards,

Anoop Kumar, Ph.D.

Academic Editor

PLOS ONE

Journal Requirements:

Reviewers' comments:

Reviewer's Responses to Questions

**Comments to the Author**

1. Is the manuscript technically sound, and do the data support the conclusions?

Reviewer #1: Partly

Reviewer #2: Partly

2. Has the statistical analysis been performed appropriately and rigorously?

Reviewer #1: Yes

Reviewer #2: Yes

3. Have the authors made all data underlying the findings in their manuscript fully available?

Reviewer #1: No

Reviewer #2: Yes

4. Is the manuscript presented in an intelligible fashion and written in standard English?

Reviewer #1: Yes

Reviewer #2: Yes

Reviewer #1: Firstly, I would like to appreciate all of the authors for their valuable contributions to the manuscript, "In silico design of novel recombinant antigens containing immunologically relevant regions of wild-type and escape mutant variants of HBsAg."

I went through the manuscript thoroughly, and the following are my insights and suggestions for further revision.

1.Please elaborate on the quality control standards used for the 617 NCBI sequences in Section 2.1. Explain why April 2021 was selected as the cut-off date and whether geographic sampling bias could have an impact on genotype frequency results. The manuscript doesn't include how sequence quality was assessed and if low quality sequences were excluded? Also, please mention if duplicates were removed to avoid bias and if any filtering was done to include all sequences were from Ethiopian-origin patients.

2.Please mention the database used, the E-value cutoff, the identity threshold, the coverage criteria, and the BLAST program (such as blastp) in Section 2.3.The term "no homology" is not defined; is it referring to a threshold of less than 30% identity over the entire sequence,or something else? The claim cannot be independently verified without this information, and the result is hard to replicate.

3.Predictions in Section 2.4 are performed using only one tool (BCEPRED). Despite being a useful tool, BCEPRED has certain drawbacks that need to be addressed:

using just one algorithm raises the possibility of false positives or negatives.

There are no parameters provided, such as window size or the propensity scale.

In the modeled 3D structures, there is no validation step, such as cross-checking. Please include these information in the manuscript.

4.I-TASSER model confidence scores (C-score, TM-score) are mentioned in passing but without explanation in Sections 2.5 and 3.3.4. Please add more information on these scores, include details on the the availability of templates for the sequences you have designed, and point out areas of low-confidence.

5.In Section 3.1, genotype frequencies and mutation percentages are presented without significance testing or statistical confidence intervals. Please mention if the proportions are robust enough ,given the size of the dataset's and potential biases.

6.The term “key diagnostic EMAVs” is used but not defined thoroughly. Please mention the criteria for considering an EMAV as “key. Please add references and citations.

7.Results (such as strong antigenicity, multiple antibody population recognition and structural soundness) are presented as conclusive in Sections 3.3.1–3.3.4. Because computational methods are uncertain in nature, please mention that the predictions need additional experimental validation.

8.Please mention accession lists of all sequences used, so other researchers can replicate the dataset's exactly.

Typographical Errors:

1.Page 17, Table 2: “MolPr obity” should be “MolProbity.”

2.Abstract and keywords: “molecular dynamic simulations” should be “molecular dynamics simulations.”

3.Page 3, line 20: “1.3 million(3)..” should have only one period: “1.3 million(3).”

Reviewer #2: Review for “In silico design of novel recombinant antigens containing immunologically relevant regions of wild-type and escape mutant variants of HBsAg”

I appreciate the hard work and the scientific approach. It’s a well written manuscript. I would like the authors to revise on the following points: I am adding my suggestions as well.

1. Since the entire study is computational, with no experimental validation (e.g., expression, purification, serological testing). The conclusions about diagnostic utility are speculative.

Authors can acknowledge this limitation clearly and, write a future plan to validate their designed constructs through:

o Cloning and expression in E. coli or mammalian cells.

o ELISA testing with sera from HBV-positive patients (wild-type and escape mutants).

o Stability/solubility checks of recombinant proteins.

2. The dataset includes only 617 sequences from Ethiopia, up to April 2021. This may not represent global or even current Ethiopian HBV diversity.

o Expand the dataset to include global sequences from GenBank, especially genotypes B, C, and E which are common in Asia and Africa.

o Incorporate newer sequences (2022–2025) to ensure up-to-date coverage.

o Perform phylogenetic comparison to demonstrate broader relevance.

3. The study focuses only on linear B-cell epitopes, whereas HBV antigens are known to have conformational epitopes critical for antibody recognition.

o Include conformational epitope prediction (e.g., DiscoTope, ElliPro).

o Consider T-cell epitope predictions to understand the immunogenic landscape more fully.

o Validate epitope conservancy against a larger HBV sequence dataset.

4. The recombinant designs (MeRPYS1–3) contain tandem repeats of the MHR and escape mutant regions linked by GGS linkers. Such designs may cause misfolding, aggregation, or reduced solubility in real expression systems.

o Evaluate protein solubility and folding using in silico tools (e.g., SOLpro, Aggrescan).

o Explore shorter constructs or mosaic antigens instead of long repeats.

o Consider codon optimization for expression in the chosen system.

5. Structural validation relies mainly on Ramachandran plots and ProSA. Docking is performed with hNTCP, which is a receptor for viral entry, not relevant for diagnostic antibody binding.

o Dock recombinant proteins against anti-HBsAg antibodies (available in PDB) instead of hNTCP, since the diagnostic purpose relates to antigen–antibody recognition.

o Complement I-TASSER results with AlphaFold2 predictions for higher structural confidence.

6. MD was done using I-MODS (normal mode analysis), which is a coarse approximation and not a full atomistic MD simulation.

o Perform atomistic MD simulations (e.g., GROMACS, AMBER) with explicit solvent models to assess stability and conformational dynamics.

o Report RMSD, RMSF, Rg, and hydrogen-bond stability metrics.

7. The conclusion that these constructs are “promising candidate antigens” is overstated without experimental evidence.

o Reframe conclusions to highlight that the study provides a computational framework for antigen design, and that further experimental validation is needed.

o Provide a more balanced discussion of limitations.

In summary:

The manuscript provides a valuable computational pipeline but lacks experimental depth. To improve, the authors should (i) broaden sequence coverage, (ii) include conformational epitope analysis, (iii) redesign docking studies with antibodies, (iv) apply more robust MD simulations, and (v) clearly state the need for laboratory validation.

**Do you want your identity to be public for this peer review?** For information about this choice, including consent withdrawal, please see our Privacy Policy

Reviewer #1: **Yes:** Annesha Chowdhury

Reviewer #2: **Yes:** Dr. Shraddha Basu

---

## [Author Response · Author response to Decision Letter 1]

28 Sep 2025

All of the responses to specific reviewer and editor comments are included in the rebuttal letter that responds to each point raised by the academic editor and reviewers and uploaded as a separate file labeled 'Response to Reviewers

---

## [Decision Letter · Decision Letter 1]

11 Nov 2025

Dear Dr. Workneh,

Thank you for submitting your manuscript to PLOS ONE. After careful consideration, we feel that it has merit but does not fully meet PLOS ONE’s publication criteria as it currently stands. Therefore, we invite you to submit a revised version of the manuscript that addresses the points raised during the review process.

We look forward to receiving your revised manuscript.

Kind regards,

Anoop Kumar, Ph.D.

Academic Editor

PLOS ONE

Journal Requirement:

Reviewers' comments:

Reviewer's Responses to Questions

**Comments to the Author**

Reviewer #2: (No Response)

2. Is the manuscript technically sound, and do the data support the conclusions?

Reviewer #2: Partly

3. Has the statistical analysis been performed appropriately and rigorously?

Reviewer #2: No

4. Have the authors made all data underlying the findings in their manuscript fully available?

Reviewer #2: Yes

5. Is the manuscript presented in an intelligible fashion and written in standard English?

Reviewer #2: No

Reviewer #2: General Assessment

This manuscript presents a computational approach to designing recombinant hepatitis B surface antigens (MeRPYS constructs) that integrate wild-type and immune escape mutant epitopes from circulating Ethiopian HBV strains. The work uses bioinformatics pipelines—epitope mapping, 3D structure modeling, molecular docking, and molecular dynamics simulations—to predict antigen–antibody interactions and structural stability.

The topic is relevant, particularly for regions with high HBV diversity where diagnostic antigen mismatches may occur. The study is methodically organized, and the authors have assembled a comprehensive set of in silico analyses. However, several areas require clarification, validation, and tempering of conclusions. At present, the manuscript remains computationally strong but biologically speculative.

Major comments:

1. Lack of Experimental Validation

The study’s conclusions about antigenicity and diagnostic potential rely entirely on in silico predictions. While computational studies are valuable, the claims of “diagnostic applicability” or “binding efficiency” cannot be substantiated without in vitro data.

• The authors should acknowledge more clearly that the findings are hypothesis-generating, not confirmatory.

• Ideally, expression and immunoreactivity testing of at least one construct (e.g., MeRPYS1) with HBV-positive and negative sera should be included.

• If laboratory validation is not feasible at this stage, a dedicated future work section should outline a feasible experimental roadmap (expression host, detection method, ELISA validation, etc.).

2. Docking Energy Interpretation

The reported docking energies (approximately −1000 kcal/mol) are unrealistically high for antigen–antibody interactions, which typically range between −6 to −15 kcal/mol. This discrepancy likely arises from the scoring scale used by ClusPro or another unnormalized energy unit.

• The authors should clarify what these values represent, provide normalized binding energies, or use additional scoring such as MM/PBSA to yield physically meaningful results.

• Without correction, these energy values risk misleading readers about the strength of antigen–antibody binding.

3. Molecular Dynamics (MD) Simulation Analysis

The MD results (RMSD, Rg, SASA, and FEL analyses) are presented, but their biological significance is not sufficiently discussed.

• Please interpret what structural stabilization implies for antigen–antibody interface robustness and potential folding quality.

• The discussion should compare fluctuations between the free antigen and the antigen–antibody complex.

• Indicate simulation length (e.g., 100 ns, 200 ns) and whether it is sufficient for convergence.

4. Immunoinformatics Prediction Limitations

The work assumes that predicted epitopes equate to immunogenic or diagnostically relevant epitopes. However, antigenicity predictions do not always correlate with actual antibody recognition due to folding constraints and post-translational modifications.

• The manuscript should temper statements that suggest these epitopes are confirmed immunogenic regions.

• Add references acknowledging prediction limitations (e.g., Jespersen et al., Front. Immunol. 2017).

5. Novelty and Positioning

The concept of recombinant HBV antigens combining wild-type and escape mutants is not entirely new. Similar constructs have been reported for diagnostic antigen optimization.

• The novelty here lies in focusing on Ethiopian HBV genotypes and escape profiles, which should be explicitly stated.

• Rephrase claims like “first design” to “region-specific computational design of HBsAg diagnostic constructs.”

6. Genotype Analysis and Sequence Statistics

The frequency comparison between genotypes A and D is mentioned but lacks details on statistical testing or visualization.

• Include the number of sequences analyzed per genotype, the source database, and statistical significance (e.g., chi-square test).

• A simple heatmap or frequency bar chart summarizing amino acid substitutions would improve clarity.

7. Expression Feasibility

Although the design incorporates linkers and tags, expression feasibility is not evaluated.

• Add codon optimization results (CAI, GC%) for a likely host (e.g., E. coli).

• Predict solubility and signal peptide presence using tools such as SOLpro or SignalP.

Minor comments:

1. Clarity of Methods:

• Specify the exact versions and parameters of each tool used (IEDB, VaxiJen, ClusPro, GROMACS, etc.).

• Define the basis for selecting epitopes—thresholds, prediction cutoffs, or ranking criteria.

2. Figures:

• Figures 5–8 (structural modeling and docking) can be combined into a single composite workflow figure.

• Figure legends are verbose and sometimes repeat information; make them more concise.

• Include residue labels for key contact sites in docking visuals.

3. Table Presentation:

• Some tables lack clear titles and units. Include binding energy units and consistent decimal formatting.

4. Statistical Reporting:

• Provide mean ± SD for numerical data where applicable.

• Report how many simulation replicates were performed.

5. References:

• Check recent studies on HBV escape mutations and diagnostic antigen design (2023–2024) to strengthen the discussion.

• Ensure all URLs or database accessions are up to date and functional.

Editorial and Language Comments:

The manuscript is generally understandable but requires polishing for grammar and flow. Common issues include missing articles, verb–subject disagreements, and repetition.

Example:

“The small hepatitis B surface antigen (HBsAg), which consists of major hydrophilic region and ‘a’ determinant region…”

Suggested Correction:

“The small hepatitis B surface antigen (HBsAg) comprises the major hydrophilic region (MHR) and the ‘a’ determinant.”

Example:

“This study is the first study to design…”

Suggested Correction:

“This study is the first to design…”

Example:

“Each blue dot represents a single amino acid residue.”

Suggested Correction:

“Each blue dot corresponds to an amino acid residue.”

Example:

“These results confirmed that the designed antigens are immunologically reactive.”

Suggested Correction:

“These results suggest that the designed antigens are potentially immunoreactive.”

Example:

“Proteins were constructed by assembling ten tandem copies of MHR segments.”

Suggested Correction:

“Proteins were constructed from ten tandem repeats of MHR segments.”

Example:

“Physiochemical”

Suggested Correction:

“Physicochemical”

Example:

“A flexible linker was added to connect two epitopes”

Suggested Correction:

“Flexible linkers (GSGSG) were introduced to maintain structural flexibility between epitopes.”

General stylistic suggestions:

• Maintain consistent use of either American or British English.

• Avoid redundancy (“HBsAg antigen” → “HBsAg”).

• Simplify long sentences—many exceed 35 words and can be split for readability.

• Ensure that all acronyms are defined upon first use.

Decision: Major Revision

While the computational analyses are detailed and relevant, the manuscript requires:

• Clarified energy interpretation and docking validation,

• Deeper discussion of MD results,

• Explicit acknowledgment of the limitations of predictive modeling,

• Clearer statistical and methodological descriptions, and

• Careful language editing.

Experimental data are not mandatory for PLOS ONE if the modeling is robust, but the conclusions must be scaled down to reflect a computational hypothesis rather than proven diagnostic efficacy.

**Do you want your identity to be public for this peer review?** For information about this choice, including consent withdrawal, please see our Privacy Policy

Reviewer #2: **Yes:** Shraddha Basu

---

## [Author Response · Author response to Decision Letter 2]

15 Dec 2025

All responses are included in the "rebuttal letter"

---

## [Decision Letter · Decision Letter 2]

29 Dec 2025

Dear Dr. Workneh,

Thank you for submitting your manuscript to PLOS ONE. After careful consideration, we feel that it has merit but does not fully meet PLOS ONE’s publication criteria as it currently stands. Therefore, we invite you to submit a revised version of the manuscript that addresses the points raised during the review process.

https://journals.plos.org/plosone/s/submission-guidelines#loc-laboratory-protocols . Additionally, PLOS ONE offers an option for publishing peer-reviewed Lab Protocol articles, which describe protocols hosted on protocols.io. Read more information on sharing protocols at https://plos.org/protocols?utm_medium=editorial-email&utm_source=authorletters&utm_campaign=protocols .

We look forward to receiving your revised manuscript.

Kind regards,

Anoop Kumar, Ph.D.

Academic Editor

PLOS One

Journal Requirements:

Reviewers' comments:

Reviewer's Responses to Questions

**Comments to the Author**

Reviewer #2: (No Response)

2. Is the manuscript technically sound, and do the data support the conclusions?

Reviewer #2: Partly

3. Has the statistical analysis been performed appropriately and rigorously?

Reviewer #2: No

4. Have the authors made all data underlying the findings in their manuscript fully available?

Reviewer #2: Yes

5. Is the manuscript presented in an intelligible fashion and written in standard English?

Reviewer #2: No

Reviewer #2: Overall Assessment

This is a well-structured, comprehensive in silico study addressing an important and regionally relevant problem: the impact of HBsAg escape mutant variants on HBV diagnosis in Ethiopia. The computational workflow is extensive, logically organized, and technically sound, covering sequence analysis, antigen design, epitope prediction, structure modeling, docking, and MD simulations.

The manuscript demonstrates strong effort, depth, and clarity of intent, and it is suitable for a journal like PLOS ONE. However, several methodological, interpretational, and presentation-related issues need to be addressed before acceptance.

Recommendation

Minor Revision

This recommendation is due to the following reasons:

• Some methodological choices need stronger justification

• There is over-interpretation of in silico results

• Several sections require language polishing and structural tightening

• Some figures/tables and claims need clarification or correction

With proper revisions, the manuscript could become publishable.

Section-by-Section Revision Suggestions

1. Abstract

Issues:

• Some statements slightly overstate conclusions from purely computational work.

• The abstract is long and could be tightened.

Suggestions:

• Replace phrases like:

o “demonstrated the probable antigen-antibody interactions”

with

o “suggested potential antigen-antibody interactions”

• Consider shortening background statistics to focus more on novelty and results.

2. Introduction

Strengths:

• Strong epidemiological context

• Clear rationale for region-specific antigen design

Issues:

• Some repetition regarding diagnostic escape mutants

• Overuse of global HBV statistics where regional context would be stronger

Suggestions:

• Reduce redundancy in paragraphs discussing EMAVs

• Clearly state what gap this study fills compared to previous in silico HBV antigen studies

• Explicitly clarify why polyclonal antibody production is the chosen downstream goal (diagnostics vs vaccines)

3. Methods

Key concerns:

a) Sequence dataset justification

• Exclusion of genotype E (KT367574) needs stronger justification, as even rare genotypes may influence diagnostic design.

• Clarify whether the dataset represents unique patients or multiple sequences per patient.

b) Statistical analysis

• Two-proportion z-test is mentioned but:

o No correction for multiple testing

o No effect size discussion

c) Epitope prediction thresholds

• Choice of:

o BCPRED cutoff

o ELIPRO score ≥ 0.5

o MHC-II percentile ≤ 0.1

should be justified with citations.

d) Molecular dynamics

• Only one MD run per complex was performed.

o This is a limitation and should be acknowledged more explicitly.

Suggestions:

• Add a limitations paragraph within Methods or Discussion explaining:

o Single-run MD simulations

o Reliance on murine MHC-II only

• Clarify reproducibility (random seeds, repeatability)

4. Results and Discussion

This is the strongest section scientifically, but also where over-interpretation occurs.

a) EMAV prevalence analysis

• Very solid

• Consider adding a supplementary table listing all EMAVs and frequencies

b) Epitope prediction

• Statements like:

“potentially induce polyclonal antibody production”

should be softened to

“may support polyclonal antibody responses”

• Clarify that epitope duplication ≠ guaranteed immunodominance

c) Structural modeling

• Well done

• However, Ramachandran plot interpretation should avoid phrases like “well-formed” and instead use “acceptable stereochemical quality”

d) Docking and MD simulations

Issues:

• Docking scores are relative, but sometimes discussed as indicators of binding strength

• RMSD values reported (<1.75 Å) seem unusually low for large antigen–antibody complexes—needs clarification:

o backbone only?

o antigen only?

• Free-energy landscape units described as “arbitrary units” without explanation

Suggestions:

• Explicitly state:

o RMSD atom selection

o Docking is qualitative, not quantitative

• Tone down claims of “strong binding” and use “stable interaction patterns”

5. Conclusion

Issues:

• Slight repetition of Results

• Some statements verge on confirmatory tone

Suggestions:

• Focus on what was achieved computationally

• End with a stronger emphasis on hypothesis generation

6. Limitations and Future Directions

Strength:

• Honest and transparent

Suggestions:

• Add:

o Limitation of murine MHC predictions for human diagnostics

o Lack of glycosylation modeling (important for HBsAg)

7. Language and Style

Common issues:

• Article misuse (“the”, “a”)

• Verb tense inconsistency

• Occasional long, dense sentences

Recommendation:

• Professional English language editing strongly recommended

• Especially for Results and Discussion

Final Decision

Rationale:

• The science is solid and meaningful

• The computational workflow is impressive

• However, claims must be strengthened, and methodological clarity improved

• Language polishing is necessary for journal standards

**Do you want your identity to be public for this peer review?** For information about this choice, including consent withdrawal, please see our Privacy Policy

Reviewer #2: **Yes:** Shraddha Basu

---

## [Author Response · Author response to Decision Letter 3]

12 Jan 2026

All information is incorporated in the rebuttal letter.

---

## [Decision Letter · Decision Letter 3]

27 Jan 2026

Dear Dr. Workneh,

Thank you for submitting your manuscript to PLOS ONE. After careful consideration, we feel that it has merit but does not fully meet PLOS ONE’s publication criteria as it currently stands. Therefore, we invite you to submit a revised version of the manuscript that addresses the points raised during the review process.

https://journals.plos.org/plosone/s/submission-guidelines#loc-laboratory-protocols . Additionally, PLOS ONE offers an option for publishing peer-reviewed Lab Protocol articles, which describe protocols hosted on protocols.io. Read more information on sharing protocols at https://plos.org/protocols?utm_medium=editorial-email&utm_source=authorletters&utm_campaign=protocols .

We look forward to receiving your revised manuscript.

Kind regards,

Anoop Kumar, Ph.D.

Academic Editor

PLOS One

**Journal Requirements:**

Reviewers' comments:

Reviewer's Responses to Questions

**Comments to the Author**

Reviewer #2: (No Response)

2. Is the manuscript technically sound, and do the data support the conclusions?

Reviewer #2: Yes

3. Has the statistical analysis been performed appropriately and rigorously?

Reviewer #2: Yes

4. Have the authors made all data underlying the findings in their manuscript fully available?

Reviewer #2: No

5. Is the manuscript presented in an intelligible fashion and written in standard English?

Reviewer #2: Yes

**Reviewer #2:** Dear Authors,

This manuscript presents a comprehensive in silico design and characterization of three recombinant HBsAg-derived antigens (MeRPYS1–3) intended for improved HBV diagnostic applications in Ethiopia. The study integrates sequence epidemiology, epitope prediction, structural modeling, molecular docking, and molecular dynamics simulations.

The work addresses a relevant public-health and diagnostic problem, particularly in regions with high HBV burden and genotype diversity. The computational pipeline is extensive and technically competent. However, the manuscript currently overinterprets in silico findings, lacks clarity in several methodological justifications, and requires significant tightening of claims to meet PLOS ONE’s standards for methodological rigor and reproducibility.

Comments (Should be addressed)

1. Overinterpretation of in silico predictions

While the authors repeatedly acknowledge the lack of experimental validation, several conclusions still go beyond what is justified by computational analyses alone.

Examples:

• Statements suggesting “stable antigen–antibody interactions” and “supporting candidacy for polyclonal antibody production” are not experimentally substantiated.

• MD simulation stability is interpreted as evidence of functional binding, which is not equivalent to biochemical affinity or immunoreactivity.

Recommendation:

Rephrase conclusions throughout the Results, Discussion, and Abstract to emphasize hypothesis-generating rather than confirmatory outcomes. Avoid language implying functional efficacy.

2. Limited novelty relative to existing in silico multi-epitope antigen studies

The manuscript claims to be the first to computationally design region-specific HBV antigens. However:

• Similar multi-epitope and mutation-inclusive HBsAg designs have been reported previously (including outside Ethiopia).

• The novelty here appears to be regional sequence prioritization, not the computational strategy itself.

Recommendation:

Clarify novelty explicitly:

• Emphasize Ethiopia-specific EMAV prevalence data

• Avoid blanket “first study” claims unless clearly justified with citations

3. Statistical analysis is correct but conceptually underused

The genotype A vs D comparison is statistically strong (z = 12.44, Δp = 50.7%), but:

• This result is not integrated mechanistically into antigen design decisions.

• It is unclear how genotype-specific mutation prevalence quantitatively informed MeRPYS1 vs MeRPYS2 composition.

Recommendation:

Explicitly link statistical findings to design rationale:

• Why do MeRPYS1 and MeRPYS2 differ?

• How does genotype D’s higher EMAV burden influence antigen architecture?

4. Epitope prediction redundancy inflates perceived immunogenicity

The repeated duplication of identical or near-identical epitopes is presented as beneficial, but:

• Epitope repetition does not guarantee improved immunogenicity

• Repetition can cause epitope masking, misfolding, or immune tolerance

The manuscript mentions this limitation briefly, but still implicitly treats duplication as advantageous.

Recommendation:

Strengthen critical discussion:

• Explicitly state that epitope duplication is speculative

• Cite studies where repetition failed or reduced immunogenicity

5. Molecular docking and MD simulation limitations are underemphasized

Concerns:

• Only one antibody structure (PDB: 6VJT) was used

• MD simulations were run once per complex, without replicas

• Binding free energy was not calculated (e.g., MM-PBSA/MM-GBSA)

Recommendation:

Either:

• Add justification for these limitations

or

• Explicitly acknowledge that docking + MD are qualitative, not quantitative

6. Diagnostic relevance remains hypothetical

The manuscript repeatedly frames the antigens as candidates for diagnostic assay development, but:

• No comparison is made to existing commercial HBsAg antigens

• No sensitivity/specificity modeling is attempted

Recommendation:

Tone down diagnostic claims or add a short section outlining:

• How these antigens would outperform current kits

• What specific diagnostic formats are envisioned (ELISA, lateral flow, etc.)

Minor Comments (Should be addressed)

1. Language and clarity

o Numerous grammatical errors (e.g., “excremental validation” → experimental validation)

o Inconsistent spacing, punctuation, and hyphenation

2. Figure density

o Some figures (especially MD plots) are excessive for PLOS ONE

o Consider moving secondary MD plots to Supporting Information

3. Methods reproducibility

o Random seed handling in MD simulations should be more clearly described

o Provide exact software versions consistently (some are missing)

4. Ethics statement

o Although sequence data are public, explicitly state that no new human data were generated

5. References

o Some citations supporting epitope prediction accuracy are outdated

o Consider adding recent benchmarking studies (2022–2024)

Overall Recommendation

Minor Revision

Rationale:

The study is technically solid, regionally relevant, and suitable for PLOS ONE if claims are appropriately restrained and methodological justifications strengthened. The manuscript currently reads closer to a computational validation paper than a diagnostic development study, and this distinction must be made explicit.

**Do you want your identity to be public for this peer review?** For information about this choice, including consent withdrawal, please see our Privacy Policy

Reviewer #2: **Yes:** Shraddha Basu

---

## [Author Response · Author response to Decision Letter 4]

16 Feb 2026

All responses are included in the rebuttal letter

---

## [Editor Report · Decision Letter 4]

20 Feb 2026

In silico design of novel recombinant antigens containing immunologically relevant regions of wild-type and escape mutant variants of HBsAg

PONE-D-25-38657R4

Dear Dr. Workneh,

We’re pleased to inform you that your manuscript has been judged scientifically suitable for publication and will be formally accepted for publication once it meets all outstanding technical requirements.

Kind regards,

Anoop Kumar, Ph.D.

Academic Editor

PLOS One
---

## [Editor Report · Acceptance letter]

PONE-D-25-38657R4

PLOS One

Dear Dr. Workneh,

I'm pleased to inform you that your manuscript has been deemed suitable for publication in PLOS One. Congratulations! Your manuscript is now being handed over to our production team.

Kind regards,

on behalf of

Dr. Anoop Kumar

Academic Editor

PLOS One